# Differentially Private Stochastic Expectation Propagation

**Margarita Vinaroz**  *margarita.vinaroz@tuebingen.mpg.de*

*University of Tübingen*
International Max Planck Research School for Intelligent Systems (IMPRS-IS)

**Mi Jung Park**  *mijungp@cs.ubc.ca*
*University of British Columbia*
CIFAR AI Chair at AMII

**Reviewed on OpenReview:** `https://openreview.net/forum?id=e5ILb2Nqst`

## Abstract

We are interested in privatizing an approximate posterior inference algorithm, called *Expectation Propagation (EP)*. EP approximates the posterior distribution by iteratively refining approximations to the *local* likelihood terms. By doing so, EP typically provides better posterior uncertainties than *variational inference (VI)* which *globally* approximates the likelihood term. However, EP needs a large memory to maintain all local approximations associated with each datapoint in the training data. To overcome this challenge, *stochastic expectation propagation (SEP)* considers a single unique local factor that captures the average effect of each likelihood term to the posterior and refines it in a way analogous to EP. In terms of privatization, SEP is more tractable than EP. It is because at each factor's refining step we fix the remaining factors, where these factors are independent of other datapoints, which is different from EP. This independence makes the sensitivity analysis straightforward. We provide a theoretical analysis of the privacy-accuracy trade-off in the posterior distributions under our method, which we call *differentially private stochastic expectation propagation (DP-SEP)*. Furthermore, we test the DP-SEP algorithm on both synthetic and real-world datasets and evaluate the quality of posterior estimates at different levels of guaranteed privacy.

## 1 Introduction

Bayesian learning provides a level of uncertainty about a model through the posterior distribution over the parameters. The posterior distribution then provides a level of uncertainty about the model's prediction through the posterior predictive distribution. Variational inference (VI) (Beal, 2003; Jordan et al., 1999) is a popular Bayesian inference method that refines a global approximation of the posterior and scales well to applications with large datasets. However, VI often underestimates the variance of the posterior and performs poorly for models with non-smooth likelihoods (Cunningham et al., 2013; Turner & Sahani, 2011).

In contrast, expectation propagation (EP) is known to provide better posterior uncertainties than VI (Minka, 2001; Opper & Winther, 2005). EP constructs the approximate posterior by iterating local computations that refine approximating factors, where each factor captures each likelihood's contribution to the posterior. With large datasets, however, using EP imposes challenges as maintaining each of the local approximates in memory is costly. Stochastic expectation propagation (SEP) (Li et al., 2015) overcomes this challenge by iteratively refining a single approximating factor that is repeated as many times as the number of datapoints that are in the dataset.

The idea behind SEP is that the unique factor captures the average effect of the likelihoods to the posterior. Employing a single approximating factor makes the algorithm suitable for large-scale datasets as it needs

to keep the global approximating factor only, as opposed to EP that needs to keep all the approximating factors in memory. While SEP is not exactly EP but approximates EP, SEP is known to provide the posterior uncertainties very close to the ones in EP.

Despite EP and SEP's advantage over VI, the standard form of the algorithms unfortunately cannot guarantee privacy for each individual in the dataset. In particular, EP and SEP enables approximate Bayesian inference under a broad class of models such as generalized linear models and mixed models. However, when these models are trained with privacy-sensitive data, the approximate posteriors can potentially leak information on the training examples.

Differential privacy (DP) (Dwork & Roth, 2014) has become the gold standard for providing privacy and is widely used in diverse applications from medicine to social science. To apply DP to EP, a difficulty arises in sensitivity analysis: at each step of the algorithm, the approximating factor that is being refined depends on the rest of the other factors where these other factors are functions of training data. Hence, the sensitivity of the approximate posterior depends not only the particular factor that is being refined but also the rest of the factors that contribute to the posterior. Due to this fact, it is challenging to obtain the nice property of sensitivity scaling with $\frac{1}{N}$, where $N$ is the number of datapoints in the training data.

On the other hand, in every SEP step it considers a single approximating factor at a time while all the other factors are fixed to the same values either at the initial step, or at the previous training step. Hence, the sensitivity analysis of the approximate posterior becomes straightforward. In particular, the natural parameters of the approximate posterior under SEP is a linear sum of those corresponding to the likelihood factors and prior. Considering that each of the approximating factors and prior parameters are norm bounded by a constant $C$ (otherwise we can clip them to have norm $C$), then the sensitivity of the natural parameters of the approximate posterior becomes proportional to $\frac{C}{N}$ (see Sec. 3).

Taken together, we summarize our contribution of this paper.

- To the best of our knowledge, we provide the first differentially-private version of the stochastic expectation propagation algorithm, called *DP-SEP*, which is scalable for large datasets and also privacy-preserving.

- We provide a theoretical analysis of the privacy-accuracy trade-off by computing the worst-case KL divergence between the private and non-private posterior distributions.

- We also provide experimental results applied to a synthetic dataset for a mixture-of-Gaussian model and several real-world datasets for a Bayesian neural network model.

In what follows, we provide background information on expectation propagation, stochastic expectation propagation and differential privacy in Sec. 2. We then describe our DP-SEP algorithm in Sec. 3. In Sec. 4, we analyze the effect of noise added to the natural parameters on the accuracy of the differentially private posterior distributions. We describe related work in Sec. 5. Finally, we demonstrate the performance of DP-SEP in relation to other posterior inference methods such as VI, EP, and SEP in Sec. 6.

## 2 Background

In the following, we describe EP and SEP algorithms, differential privacy and its properties that we will use to develope our algorithm in Sec. 3.

### 2.1 Expectation propagation (EP)

We consider a dataset $\mathcal{D} = \{\boldsymbol{x}_n\}_{n=1}^N$ containing $N$ i.i.d samples. Given the dataset, we pick a model parameterized by $\boldsymbol{\theta}$. We denote the likelihood of a datapoint $\boldsymbol{x}_n$ given the model by $p(\boldsymbol{x}_n|\boldsymbol{\theta})$ and the prior distribution over the parameters by $p_0(\boldsymbol{\theta})$. The true (intractable) posterior distribution is proportional to the

---

**Algorithm 1** EP

---

1: Choose a factor $f_n$ to refine
2: Compute the cavity distribution
   $q_{-n}(\boldsymbol{\theta}) \propto q(\boldsymbol{\theta})/f_n(\boldsymbol{\theta})$
3: Compute the tilted distribution
   $\tilde{p}_n(\boldsymbol{\theta}) \propto p(\boldsymbol{x}_n|\boldsymbol{\theta})q_{-n}(\boldsymbol{\theta})$
4: Moment matching
   $f_n(\boldsymbol{\theta}) \leftarrow \text{proj}[\tilde{p}_n(\boldsymbol{\theta})]/q_{-n}(\boldsymbol{\theta})$
5: Inclusion
   $q(\boldsymbol{\theta}) \leftarrow q_{-n}(\boldsymbol{\theta})f_n(\boldsymbol{\theta})$

---

**Algorithm 2** SEP

---

1: Choose a datapoint $\boldsymbol{x}_n \sim \mathcal{D}$
2: Compute the cavity distribution
   $q_{-1}(\boldsymbol{\theta}) \propto q(\boldsymbol{\theta})/f(\boldsymbol{\theta})$
3: Compute the tilted distribution
   $\tilde{p}_n(\boldsymbol{\theta}) \propto p(\boldsymbol{x}_n|\boldsymbol{\theta})q_{-1}(\boldsymbol{\theta})$
4: Moment matching
   $f_n(\boldsymbol{\theta}) \leftarrow \text{proj}[\tilde{p}_n(\boldsymbol{\theta})]/q_{-1}(\boldsymbol{\theta})$
5: Implicit update
   $f(\boldsymbol{\theta}) \leftarrow f(\boldsymbol{\theta})^{1-\frac{\gamma}{N}} f_n(\boldsymbol{\theta})^{\frac{\gamma}{N}}$
6: Inclusion
   $q(\boldsymbol{\theta}) \leftarrow q_{-1}(\boldsymbol{\theta})f(\boldsymbol{\theta})$

---

product of the prior and the likelihood, given by:

$$p(\boldsymbol{\theta}|\mathcal{D}) \propto p_0(\boldsymbol{\theta}) \prod_{n=1}^{N} p(\boldsymbol{x}_n|\boldsymbol{\theta}). \tag{1}$$

EP is an iterative algorithm that produces a simpler and tractable approximate posterior distribution, $q(\boldsymbol{\theta})$, by refining the approximating factors $f_n(\boldsymbol{\theta})$ associated with each datapoint, given by:

$$p(\boldsymbol{\theta}|\mathcal{D}) \approx q(\boldsymbol{\theta}) \propto p_0(\boldsymbol{\theta}) \prod_{n=1}^{N} f_n(\boldsymbol{\theta}). \tag{2}$$

EP refines these factors iteratively, as shown in Algorithm 1. Firstly, we initialize the approximating factors and form the cavity distribution $q_{-n}(\boldsymbol{\theta})$ by taking the n-th approximating factor out from the approximated posterior (i.e $q_{-n}(\boldsymbol{\theta}) \propto q(\boldsymbol{\theta})/f_n(\boldsymbol{\theta})$). Secondly, we compute the tilted distribution, $\tilde{p}_n(\boldsymbol{\theta})$, by including the corresponding likelihood term to the cavity distribution: $\tilde{p}_n(\boldsymbol{\theta}) \propto q_{-n}(\boldsymbol{\theta})p(\boldsymbol{x}_n|\boldsymbol{\theta})$. Thirdly, we update the approximating factor by minimizing the Kullback-Leibler (KL) divergence between the tilted distribution and $q_{-n}(\boldsymbol{\theta})f_n(\boldsymbol{\theta})$ to capture the likelihood term's contribution to the posterior. When the approximate distribution belongs to the exponential family, the KL minimization reduces to moment matching (Amari & Nagaoka, 2000), denoted by: $f_n(\boldsymbol{\theta}) \leftarrow \text{proj}[\tilde{p}(\boldsymbol{\theta})]/q_{-n}(\boldsymbol{\theta})$. Finally, we add the refined factor $f_n(\boldsymbol{\theta})$ to the approximate posterior in the inclusion step. We repeat this process until some convergence criterion is satisfied.

## 2.2 Stochastic EP (SEP)

A major difference between EP and SEP is that SEP constructs an approximate posterior, $q(\boldsymbol{\theta})$, by iteratively refining $N$ copies of *a unique factor*, $f(\boldsymbol{\theta})$, such that $\prod_{n=1}^{N} p(\boldsymbol{x}_n|\boldsymbol{\theta}) \approx f(\boldsymbol{\theta})^N$. The intuition behind SEP is that the approximating factor captures the average effect of the likelihood on the posterior distribution, since the updates are performed analogously to EP.

Similar to EP, as shown in Algorithm 2, we start by initializing the approximating factor and computing the cavity distribution by removing the factor from the approximate posterior: $q_{-1}(\boldsymbol{\theta}) \propto q(\boldsymbol{\theta})/f(\boldsymbol{\theta})$. Note that unlike EP, where the cavity distribution involve a datapoint's index $n$, this cavity distribution is independent of a data index, as we obtain it by removing one datapoint's average worth – determined by $f(\boldsymbol{\theta})$ – from the posterior distribution.

We then calculate the tilted distribution in the same way as in EP by $\tilde{p}_n(\boldsymbol{\theta}) \propto q_{-1}(\boldsymbol{\theta})p(\boldsymbol{x}_n|\boldsymbol{\theta})$. In the third step, we minimize the KL-divergence between the tilted distribution and $q_{-1}(\boldsymbol{\theta})f_n(\boldsymbol{\theta})$ to find an intermediate factor, $f_n(\boldsymbol{\theta})$. In the last step, we update the factor with a damping rate $\gamma/N$: $f(\boldsymbol{\theta}) \leftarrow f(\boldsymbol{\theta})^{1-\gamma/N}f_n(\boldsymbol{\theta})^{\gamma/N}$. A common choice for the damping factor is $1/N$ because it can be seen as minimizing the KL divergence between the tilted distribution and $p_0(\boldsymbol{\theta})f(\boldsymbol{\theta})^N$.

In the last step of the algorithm, we include the refined factor in the approximate posterior. We repeat these steps until convergence. The SEP algorithm reduces the storage requirement compared to EP as it only maintains the global approximation, where the following holds:

$$f(\boldsymbol{\theta}) \propto (q(\boldsymbol{\theta})/p_0(\boldsymbol{\theta}))^{\frac{1}{N}} \tag{3}$$

$$q_{-1}(\boldsymbol{\theta}) \propto q(\boldsymbol{\theta})^{1-\frac{1}{N}} p_0(\boldsymbol{\theta})^{\frac{1}{N}} \tag{4}$$

## 2.3   Differential privacy

Given privacy parameters $\epsilon \geq 0, \delta \geq 0$ randomized algorithm, $\mathcal{M}$, is said to be $(\epsilon, \delta)$-DP (Dwork & Roth, 2014) if for all possible sets of mechanism's outputs $S$ and for all neighboring datasets $\mathcal{D}, \mathcal{D}'$ differing in an only single entry $(d(\mathcal{D}, \mathcal{D}') \leq 1)$, the following inequality holds:

$$\Pr[\mathcal{M}(\mathcal{D}) \in S] \leq e^\epsilon \cdot \Pr[\mathcal{M}(\mathcal{D}') \in S] + \delta.$$

The definition states that the amount of information revealed by a randomized algorithm about any individual's participation is limited. And the amount is determined by $\epsilon$ and $\delta$.

A common way of constructing a differentially private algorithm is to add a calibrated amount of noise to an output of the algorithm. Suppose we want to privatize a function $h : \mathcal{D} \to \mathbb{R}^d$, which takes a dataset as an input and output a $d$-dimensional real-valued vector. The *Gaussian mechanism* adds noise such that the output of the mechanism is given by $\tilde{h}(\mathcal{D}) = h(\mathcal{D}) + \mathbf{n}$, where $\mathbf{n} \sim \mathcal{N}(0, \sigma^2 \Delta_h^2 \mathbf{I}_d)$. Here, the noise scale depends on the *global sensitivity* (Dwork et al., 2006a) of the function $h$ denoted by $\Delta_h$, which is defined as the $L_2$-norm $\|h(\mathcal{D}) - h(\mathcal{D}')\|_2$ where $\mathcal{D}, \mathcal{D}'$ are neighboring datasets differing in an only single entry. The Gaussian mechanism is $(\epsilon, \delta)-$DP and $\sigma$ is a function that depends on $\epsilon, \delta$. For a single application of the mechanism, $\sigma \geq \sqrt{2\log(1.25/\delta)}/\epsilon$ for $\epsilon \geq 1$. In our algorithm, we will use the Gaussian mechanism to privatize the approximate posterior distribution.

There are two important properties of differential privacy. The first one is *post-processing invariance* (Dwork et al., 2006b), which states that the composition of any data-independent mapping with an $(\epsilon, \delta)$-DP algorithm is also $(\epsilon, \delta)$-DP. What this means in our context is that no analysis on our privatized approximate posterior can yield more information about the training data than what our choice of $\epsilon$ and $\delta$ allows.

The second property is *composability* Dwork et al. (2006a), which states that the strength of privacy guarantee degrades in a measurable way with repeated use of the training data. In this work, we use the subsampled RDP composition (Wang et al., 2019) as a composition technique, as it provides tight bounds on the cumulative privacy loss when we subsample datapoints from the training data. For this, we use the auto-dp package (https://github.com/yuxiangw/autodp) to compute the privacy parameter $\sigma$ given our choice of $\epsilon, \delta$ values and the number of times we access data while running our algorithm.

---

**Algorithm 3** DP-SEP

---

**Require:** Dataset $\mathcal{D}$. Initial natural parameters for approximating factor $\|\boldsymbol{\theta}_f\|_2 \leq C$ and those for the prior $\|\boldsymbol{\theta}_0\|_2 \leq C$, damping value $\gamma$, and the privacy parameter $\sigma$.

**Ensure:** $(\epsilon, \delta)$-DP natural parameters of the approximate posterior

1: **for** $t = 1, \ldots, T$ **do**
2:     **for** $n \in \{1, \ldots, N\}$, uniformly random without replacement **do**
3:         Choose a datapoint $\boldsymbol{x}_n \sim \mathcal{D}$
4:         Compute cavity distribution: $q_{-1}(\boldsymbol{\theta}) \propto q(\boldsymbol{\theta})/f(\boldsymbol{\theta})$
5:         Compute tilted distribution: $\tilde{p}_n(\boldsymbol{\theta}) \propto q_{-1}(\boldsymbol{\theta})p(\boldsymbol{x}_n|\boldsymbol{\theta})$
6:         Moment matching: $f_n(\boldsymbol{\theta}) \leftarrow \text{proj}[\tilde{p}_n(\boldsymbol{\theta})]/q_{-1}(\boldsymbol{\theta})$
7:         Clip the norm of the natural parameters: $\|\boldsymbol{\theta}_{f_n}\|_2 \leq C$
8:         Update the approximate posterior: $q^{\text{new}}(\boldsymbol{\theta}) \leftarrow f_n(\boldsymbol{\theta})^{\frac{\gamma}{N}} f(\boldsymbol{\theta})^{1-\frac{\gamma}{N}} q_{-1}(\boldsymbol{\theta})$
9:         Add noise to natural parameters: $\tilde{\boldsymbol{\theta}}_{\text{new}} = \boldsymbol{\theta}_{\text{new}} + \mathbf{n}$ where $\mathbf{n} \sim \mathcal{N}(0, \sigma^2 \Delta_{\boldsymbol{\theta}_{\text{new}}}^2 I)$
10:       Post-process natural parameters corresponding to covariance to ensure positive definiteness
11:       Update the approximating factor: $f(\boldsymbol{\theta}) \propto \left(q^{\text{new}}(\tilde{\boldsymbol{\theta}}_{\text{new}})/p_0(\boldsymbol{\theta})\right)^{\frac{1}{N}}$.
12:       Clip the norm of the natural parameters: $\|\boldsymbol{\theta}_f\|_2 \leq C$
13:     **end for**
14: **end for**

---

## 3 Our algorithm: DP-SEP

| | |
|---|---|
| $\mu$ | Mean parameter of a Gaussian distribution |
| $\Sigma$ | Covariance matrix of a Gaussian distribution |
| $\eta$ | Natural parameter for the mean |
| $\Lambda$ | Natural parameter for the covariance matrix |
| $\sigma^2$ | Noise variance of the Gaussian distribution |
| $\sigma_1^2$ | Noise variance of the Gaussian distribution for $\eta$ |
| $\sigma_2^2$ | Noise variance of the Gaussian distribution for $\Lambda$ |
| $\mathbb{E}_{\mathbf{E}}$ | Expectation with respect to the variable $\mathbf{E}$ |
| $A^{-1}$ | Inverse of a matrix A |
| $A^{\top}$ | Transpose of a matrix A |
| $Tr[A]$ | Trace of a matrix A |
| $|A|$ | Determinant of a matrix A |
| $\lambda_i(A)$ | i-th eigenvalue of a matrix A |
| $\lambda_{min}(A)$ | Minimum eigenvalue of a matrix A |
| $\lambda_{max}(A)$ | Maximum eigenvalue of a matrix A |

In this section, we introduce our proposed algorithm, which we call *differentially private stochastic expectation propagation (DP-SEP)*. The algorithm outputs differentially private approximate posterior distributions by noising up the natural parameters.

### 3.1 Outline of DP-SEP

**Initialization:** As shown in Algorithm 3, we first initialize the approximating factor, $f(\boldsymbol{\theta})$, such that the norm of its natural parameters $\boldsymbol{\theta}_f$ and prior natural parameters $\boldsymbol{\theta}_0$ are bounded by a constant $C$ (i.e. $\|\boldsymbol{\theta}_f\|_2 \leq C$, $\|\boldsymbol{\theta}_0\|_2 \leq C$). The norm clipping applied to each natural parameter becomes instrumental in computing the sensitivity of the natural parameters for the global approximate posterior $q(\boldsymbol{\theta})$, which is required in the later step in the algorithm. By construction, each local factor $f_n$ and the approximating factor $f$ have its own natural parameters $\boldsymbol{\theta}_{f_n}$ and $\boldsymbol{\theta}_f$, respectively. When the exponential distributions have bounded domain, the natural parameters are also norm bounded. However, when they are not norm bounded, e.g., the Gaussian distribution has an unbounded domain, we choose to clip the norm of $\boldsymbol{\theta}_{f_n}$ and $\boldsymbol{\theta}_f$ so that the natural parameters for the global approximate posterior have a limited sensitivity.

**Step 1 − 6:** The same as in the SEP algorithm in Algorithm 2, at each run of the DP-SEP algorithm, we first subsample one datapoint uniformly without replacement from the dataset, $\boldsymbol{x}_n \in \mathcal{D}$, then compute the cavity distribution $q_{-1}(\boldsymbol{\theta})$, the tilted distribution $\tilde{p}_n(\boldsymbol{\theta})$ and the intermediate factor $f_n(\boldsymbol{\theta})$ for $\boldsymbol{x}_n$, followed by the *moment matching* step.

**Step 7 − 8:** Once $f_n(\boldsymbol{\theta})$ is computed, we need to ensure that its natural parameters, $\boldsymbol{\theta}_{f_n}$, are also norm bounded by $C$ (i.e $\|\boldsymbol{\theta}_{f_n}\|_2 \leq C$). Then, the algorithm updates the natural parameters of the approximate posterior by making a partial update of the approximating factor and the cavity distribution according to the pre-selected damping value: $q^{\text{new}}(\boldsymbol{\theta}) \leftarrow f_n(\boldsymbol{\theta})^{\frac{\gamma}{N}} f(\boldsymbol{\theta})^{1-\frac{\gamma}{N}} q_{-1}(\boldsymbol{\theta})$ .

As the approximating distribution is in the exponential family, we can express the approximate posterior natural parameters, $\boldsymbol{\theta}$, as a linear combination of the natural parameters of the approximating factor and the prior (i.e. $\boldsymbol{\theta} = N\boldsymbol{\theta}_f + \boldsymbol{\theta}_0$). From this together with the damping value, we arrive at the following definition:

$$\boldsymbol{\theta}_{\text{new}} = \frac{\gamma}{N}\boldsymbol{\theta}_{f_n} + \left(N - \frac{\gamma}{N}\right)\boldsymbol{\theta}_f + \boldsymbol{\theta}_0. \tag{5}$$

**Step 9 − 11:** In the next step, we privatize the natural parameters $\boldsymbol{\theta}_{\text{new}}$ by adding the Gaussian noise with the sensitivity $\Delta_{\boldsymbol{\theta}_{\text{new}}} = \frac{2\gamma C}{N}$ (See Prop. 1). After adding Gaussian noise, the perturbed covariance natural parameter might not be positive definite. In such case, as a post-processing step, we project the negative eigenvalues to small positive values to maintain positive definiteness of the natural parameters corresponding to the covariance matrix, following (Park et al., 2017). This step does not change the level of privacy guarantee of the natural parameters after the projection, as differential privacy is post-processing invariant. Finally, in the last step, we update the unique approximating factor, $f(\boldsymbol{\theta})$, by eq. 3, using the new privatized approximate posterior denoted by $q^{\text{new}}(\tilde{\boldsymbol{\theta}}_{\text{new}})$. The updated natural parameters of the approximating factor can be then easily computed by the following expression: $\boldsymbol{\theta}_f = (\tilde{\boldsymbol{\theta}}_{\text{new}} - \boldsymbol{\theta}_0)/N$. Once we update the natural parameters for the approximating factor, we ensure that its norm is also bounded by $C$.

### 3.2 Privacy analysis

We use the subsampled Gaussian mechanism together with the analytic moments accountant for computing the total privacy loss incurred in our algorithm. Hence, we input a chosen privacy level $\epsilon, \delta$, the number of repetitions $T$, the number of datapoints $N$ and the clipping norm $C$ to the auto-dp package by (Wang et al., 2019), which returns the corresponding privacy parameter $\sigma$.

The following propositions state that (1) the sensitivity of the natural parameters is $\frac{2\gamma C}{N}$ and (2) the resulting algorithm is differentially private.

**Proposition 1.** *The sensitivity of the natural parameters, $\boldsymbol{\theta}_{new}$, in Algorithm 3 is given by $\Delta_{\boldsymbol{\theta}_{new}} = \frac{2\gamma C}{N}$.*

*Proof.* Consider two neighboring databases, $\mathcal{D}, \mathcal{D}'$ differing by an entry $n$, and same initial values for $\boldsymbol{\theta}_f, \boldsymbol{\theta}_0$:

$$
\begin{aligned}
\Delta_{\boldsymbol{\theta}_{\text{new}}} &= \max_{D,D'} \|\boldsymbol{\theta}_{\text{new}} - \boldsymbol{\theta}'_{\text{new}}\|_2 \\
&= \max_{D,D'} \| \left(\tfrac{\gamma}{N}\boldsymbol{\theta}_{f_n} + \left(N - \tfrac{\gamma}{N}\right)\boldsymbol{\theta}_f + \boldsymbol{\theta}_0\right) \\
&\quad - \left(\tfrac{\gamma}{N}\boldsymbol{\theta}'_{f_n} + \left(N - \tfrac{\gamma}{N}\right)\boldsymbol{\theta}_f + \boldsymbol{\theta}_0\right) \|_2, \text{by eq. 5} \\
&= \tfrac{\gamma}{N} \max_{D,D'} \|\boldsymbol{\theta}_{f_n} - \boldsymbol{\theta}'_{f_n}\|_2, \\
&\leq \tfrac{2\gamma}{N} \max_{D,D'} \|\boldsymbol{\theta}_{f_n}\|_2, \text{ due to the triangle inequality} \\
&= \tfrac{2C\gamma}{N}, \text{ due to the norm clipping on natural parameters.}
\end{aligned}
$$

□

**Proposition 2.** *The DP-SEP algorithm produces $(\epsilon, \delta)$-DP approximate posterior distributions.*

*Proof.* Due to the Gaussian mechanism, the natural parameters after each perturbation are DP. By composing these with the subsampled RDP composition (Wang et al., 2019), the final natural parameters are $(\epsilon, \delta)$-DP, where the exact relationship between $(\epsilon, \delta)$, $T$ (how many repetitions SEP runs), $N$ (how many datapoints a dataset has), and $\sigma$ (the privacy parameter) follows the analysis of (Wang et al., 2019). $\square$

### 3.3 A few thoughts on the algorithm

**Is this DP-SGD applied to DP-SEP?** Clipping then adding Gaussian noise seems to indicate that this algorithm is simply DP-SGD. However, there are two subtle but important differences between DP-SEP and DP-SGD. First, DP-SGD computes the gradients on the datapoints in a batch then clip the gradients. However, DP-SEP computes the global posterior distribution, which is a concatenation of all the factors associated with all the data points in the training data, not just in a selected mini-batch of the data. Hence, the sensitivity is on the order of $1/N$ where $N$ is the number of datapoints in the training data, while the sensitivity of DP-SGD is on the order of $1/B$ where $B$ is the size of the mini-batch. Hence, DP-SEP can significantly reduce the amount of noise in each privatization step, compared to DP-SGD, as typically $N \gg B$.

Second, in DP-SGD, the reason clipping the gradients is because we simply do not know how much one datapoint's difference in the neighbouring datasets would change the gradient values (that are computed on a selected mini-batch). On the other hand, in our case we do know the amount of change in the natural parameters of the global approximate posterior, as the natural parameters of the global approximate posterior are the sum of those of each factor associated with the datapoints and that of the prior. However, for the distribution with an unbounded domain, the natural parameters also have an unbounded domain. This is the reason we clip the norm of the natural parameters for the approximating factor and the local factor.

**Choice of clipping norm.** In our algorithm, we treat the clipping norm threshold $C$ as a hyperparameter, as in many other cases of DP algorithms (e.g., (Abadi et al., 2016)). When setting $C$ to a small value, the sensitivity gets also smaller which is good in terms of the amount of noise to be added. However, the small clipping norm can drastically discard information encoded in the natural parameters after the clipping procedure. Setting $C$ to a large value results in a high noise variance. However, most of the information in the natural parameters will be kept after clipping. Hence, finding the right value for the clipping norm is essential to keep the signal-to-noise ratio high. The optimal value for the clipping norm depends on the distribution of the target variables that we apply the clipping procedure. Hence, no one solution fits all.

While privacy analysis for hyperparameter tuning is an active research area (Abadi et al., 2016; Jälkö et al., 2017; Andrew et al., 2021; Kurakin et al., 2022), in this paper, we assume selecting the clipping norm does not incur privacy loss, while incorporating this aspect is an interesting research question for future work.

**Clipping as a form of regularizion** An interesting finding when applying clipping in SEP is that the clipping procedure to the natural parameters itself improves the performance of SEP, as shown in our experiments Sec. 6. It is widely known that EP frequently encounters numerical instabilities, e.g., due to the accumulation of numerical errors in local updates over the course of training, which hinders the algorithm to converge properly. SEP, on the other hand, seems to be superior to EP in terms of numerical stability, as it updates one factor that represents the average contribution of all factors to the posterior in every training step and thus numerical errors seem to accumulate slower than those in EP. From our experience, using clipping to SEP with a well-chosen clipping threshold further improves its stability, resulting in better convergence. We conjecture this is because applying the clipping procedure to the natural parameters helps avoiding any undesirable jumps in the search space, and thus effectively reduces the search space on which the algorithm focuses.

## 4 Quantitatively analysis: effect of noise

Here, we would like to analyze the effect of noise added to SEP. In particular, we are interested in analyzing the distance between the posterior distributions, where one is the posterior distribution obtained by SEP (i.e., non-DP) and the other is the posterior distribution obtained by DP-SEP. As a distance metric, we use the KL divergence between them. Thm. 4.1 formally states the effect of noise for privacy on the accuracy of

the posterior. For simplicity, we assume the posterior distribution is $d$-dimensional multivariate Gaussian. We also assume the posterior distributions between SEP and DP-SEP are compared at $T = 1, n = 1$. We denote the posterior distribution of DP-SEP (Algorithm 3) by $p := \mathcal{N}(\boldsymbol{\mu}_p, \Sigma_p)$ and the posterior distribution of SEP (Algorithm 2) by $q := \mathcal{N}(\boldsymbol{\mu}_q, \Sigma_q)$.

Before introducing the theorem, we first define the two quantities we release in every DP-SEP step.

**Definition 4.1.** *We express the first natural parameters obtained by DP-SEP as:*

$$\boldsymbol{\eta}_p = \boldsymbol{\eta}_q + \mathbf{e}$$

*where $\eta_q = \Sigma_q^{-1} \boldsymbol{\mu}_q$ is the first natural parameters obtained by SEP and $\mathbf{e}$ is iid drawn from $\mathcal{N}(0, \sigma_1^2)$.*

**Definition 4.2.** *By following Step 10 in Algorithm 3, we express the second natural parameters obtained by DP-SEP as:*

$$\Lambda_p = \Lambda_q + \mathbf{E} + A \tag{6}$$

*where $\Lambda_q = \Sigma_q^{-1}$ is the second natural parameters by SEP, $\mathbf{E}$ is a symmetric matrix where the upper triangular entries are iid drawn from $\mathcal{N}(0, \sigma_2^2)$ and the lower triangular are copied from the upper half. We construct a diagonal matrix $A$:*

$$A = \begin{cases} (-\lambda_{min}(\Lambda_q + E) + \rho)I, & when \ \sigma_2 \neq 0 \\ 0, & otherwise \end{cases} \tag{7}$$

*to ensure the resulting $\Lambda_p$ to be positive definite by taking the minimum eigenvalue of $\Lambda_q + \mathbf{E}$ and adding a small positive constant $\rho$ such that $\lambda_{min}(\Lambda_q + \mathbf{E} + A) = \rho$. Note that when there is no noise added to the covariance, we set $A = 0$, as the role of $A$ is to keep the posterior covariance to be positive definite and there is no need to add this term any longer since $\Lambda_q$ is already positive definite. By definition of $A$, setting $A = 0$ makes $\rho = \lambda_{\min}(\Lambda_q)$.*

**Theorem 4.1** (Privacy-accuracy trade-off given posteriors by Algorithm 2 and Algorithm 3)**.** *Following Def. 4.1 and Def. 4.2, with probability at least $1 - d \exp\left(\frac{-M^2}{2v(\mathbf{E})}\right)$ for any $M > 0$, the expected KL divergence between SEP and DP-SEP posterior distributions is bounded by:*

$$
\begin{aligned}
&\mathbb{E}_{\mathbf{e}}\mathbb{E}_{\mathbf{E}}(D_{kl}[q||p]) \\
&\leq \frac{1}{2}\Bigg[ Tr((-\lambda_{min}(\Lambda_q) + \sqrt{2v(\mathbf{E})\log(d)} + \rho)\Lambda_q^{-1}) \\
&+ \sigma_1^2 \lambda_{max}(\Sigma_q) \sum_{i=1}^{d} \left( \frac{h + H - 1 + \frac{\lambda_{min}(\Lambda_q) + \sqrt{2v(\mathbf{E})\log(d)} - \rho}{\lambda_i(\Lambda_q)}}{hH} \right) \\
&+ \sum_{i=1}^{d} b_i^2 \left( \frac{h + H - 1 + \frac{\lambda_{min}(\Lambda_q) + \sqrt{2v(\mathbf{E})\log(d)} - \rho}{\lambda_i(\Lambda_q)}}{hH} \right) - \eta_q^\top \Lambda_q^{-1} \eta_q \\
&+ \mathbf{a}\mathbf{a}^\top \left( -\lambda_{min}(\Lambda_q) + \sqrt{2v(\mathbf{E})\log(d)} + \rho \right) + \sum_{i=1}^{d} \log \left( \frac{h + H - 1 + \frac{\lambda_{min}(\Lambda_q) + \sqrt{2v(\mathbf{E})\log(d)} - \rho}{\lambda_i(\Lambda_q)}}{hH} \right) \Bigg]
\end{aligned}
\tag{8}
$$

*where $b_i := (\Lambda_q^{\frac{-1}{2}} \eta_q)_i$, $\mathbf{a} := \Lambda_q^{-1} \eta_q$, $h = 1 + \frac{-2M - \lambda_{min}(\Lambda_q) + \rho}{\lambda_i(\Lambda_q)}$ and $H = 1 + \frac{2M - \lambda_{min}(\Lambda_q) + \rho}{\lambda_i(\Lambda_q)}$. The matrix variance statistic is denoted by $v(E) = \|\sum_k \sigma_2^2 B_k B_k^T\|$, where the norm is spectral norm and $\mathbf{E} := \sum_{k=1}^{\frac{d(d+1)}{2}} \sigma_2 \gamma_k B_k$ where $\gamma_k$ is a standard normal Gaussian random variable and $B_k$ is a finite sequence of fixed Hermitian (symmetric) matrices. $\lambda_{max}(\Lambda_q)$ denotes the largest eigenvalue of $\Lambda_q$.*

See Sec. B in Appendix for detailed proof. The rough proof sketch is as follows. We first consider the closed-form KL divergence between the private and non-private posterior distributions with respect to the

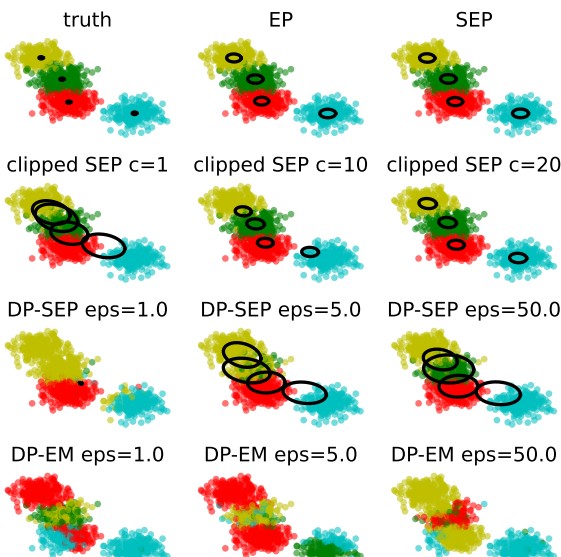

Figure 1: Posterior approximation for the mean of the Gaussian components. Black rings indicate 98 % confidence level. The coloured dots indicate the true label (top left) or the inferred cluster assignments (the rest). The top row shows EP (middle) and SEP (right). The second row shows the effect of clipping on the posterior estimate as a function of the clipping threshold $C$. The third row shows the labels for DP-SEP with $\delta = 10^{-5}$ and at $\epsilon = 1, 5, 50$. The bottom row shows the the labels for DP-EM at the same levels of privacy. Black rings for DP-EM are not shown as they appear outside the range of the plot.

two Gaussian noise distributions. Computing the expectation with respect to $\mathcal{N}(0, \sigma_1^2)$ is straightforward. However, computing the expectation with respect to $\mathcal{N}(0, \sigma_2^2)$ is more involved and we need to take into account the minimum and maximum eigenvalues of the second natural parameters (corresponding to the inverse covariance matrix) to find the desired upper bound. To do this, we use the concentration bound for Gaussian random matrices.

A few things are worth noting. Recall that each noise variance $\sigma_1 = \sigma \frac{2C}{N}$ and $\sigma_2 = \sigma \frac{2C}{N}$ (for a damping rate 1) contains the sensitivity of the natural parameters as well as the privacy parameter $\sigma$ which is a function of $\epsilon, \delta$. Also, notice $\sigma_2$ appears in $v(\mathbf{E})$ in the upper bound. This indicates that the divergence between the private posterior and non-private posterior distributions scales with $1/N$ with fixed $\epsilon, \delta$.

In the limit of infinite amounts of data, $\lim_{N \to \infty} \sigma_1 = 0$ and $\lim_{N \to \infty} \sigma_2 = 0$. At this limit, the eigenvalues of the noise matrix also $\lim_{\sigma_2 \to 0} \lambda_i(\mathbf{E}) = 0$ for all $i$. And, for any $M \geq 0$, the probability $P[\lambda_{max}(\mathbf{E}) \leq M] \leq 1 - d \exp(-\frac{M^2}{2v(\mathbf{E})}) \to 1$, as $\lim_{\sigma_2 \to 0} v(\mathbf{E}) = 0$. In this case, the gap between the private and non-private posterior distributions becomes closed. See Sec. B.1 in Appendix for the detailed discussion of the bound when $N \mapsto \infty$.

Thm. 4.1 does not take explicitly into account the clipped threshold for the natural parameters. Although, it can be easily taken into account by considering scaling down the $q$ natural parameters by $1/\max(1, \|\cdot\|_F/C)$ inside the $p$ natural parameters definition. These clipping thresholds are not functions of the Gaussian noise addition, and thus, do not affect computing the expectations in the upper bound. For curious readers, we also provide the KL divergence between the posterior distribution by the clipped version of SEP and that by SEP in Sec. A in Appendix.

For $T > 1$ or $n > 1$, the analysis we made here does not hold, due to the nested structure of SEP updates. However, our analysis can be interpreted as the discrepancy between these two (SEP and DP-SEP) algorithms at a single updating step given the same values given from the previous step.

Table 1: Accuracy of the posterior distribution (Mixture-of-Gaussian with Synthetic data)

| Method | F-norm on mean | F-norm on covariance | average F-norm |
|---|---|---|---|
| SEP ($\epsilon = \infty$) | 0.0020 | 0.0004 | 0.0012 |
| SEP-clipped C=20 | 0.0263 | 0.0004 | 0.0134 |
| SEP-clipped C=10 | 1.4950 | 0.0005 | 0.7477 |
| SEP-clipped C=1 | 2.2065 | 0.0459 | 1.1262 |
| DP-EM ($\epsilon = 50$) | 5.3769 | 1.3189 | 3.3479 |
| DP-EM ($\epsilon = 5$) | 10.4548 | 3.7573 | 7.1060 |
| DP-EM ($\epsilon = 1$) | 7.5166 | 57.8665 | 32.6916 |
| DP-SEP ($\epsilon = 50$) | 2.2411 | 0.0358 | 1.1385 |
| DP-SEP ($\epsilon = 5$) | 12.1623 | 1.0655 | 6.6139 |
| DP-SEP ($\epsilon = 1$) | 82.9746 | 5.0777 | 44.0262 |

## 5  Related Work

To the best of our knowledge, no prior work on differentially private expectation propagation or stochastic expectation propagation exits in the literature.

Remotely related work would be differentially private versions of Bayesian inference methods. This line of research started from (Dimitrakakis et al., 2014), which showed Bayesian posterior sampling becomes differentially private with a mild condition on the log likelihood. Then many other differentially private Bayesian inference methods appeared in the literature, which include posterior sampling (e.g., (Wang et al., 2015; Foulds et al., 2016; Zhang et al., 2016; Li et al., 2019)), variational inference (Park et al., 2020; Jälkö et al., 2017), and inference for generalized linear models (Kulkarni et al., 2021). In this paper, we compare the performance of our method to that of differentialy private VI (Jälkö et al., 2017) and differentially private expectation maximization (Park et al., 2017).

## 6  Experiments

The purpose of this section is to evaluate the performance of DP-SEP on different tasks and datasets. First, we consider a Mixture of Gaussians for clustering problem on a synthetic dataset and test DP-SEP at different levels of privacy guarantees.

In the second experiment, we consider a Bayesian neural network model for regression tasks and quantitatively compare our algorithm with other existing non-private methods for Bayesian inference. Our code is available at: `https://github.com/mvinaroz/DP-SEP`

### 6.1  Mixture of Gaussians for clustering

In this section, we consider a Mixture of Gaussian model for clustering using synthetic data. Following (Li et al., 2015), we generate a synthetic dataset containing $N = 1000$ datapoints drawn from $J = 4$ Gaussians of 4-dimensional inputs, where each mean is sampled from a Gaussian distribution $p(\boldsymbol{\mu}_j) = \mathcal{N}(\boldsymbol{\mu}; \mathbf{m}, I)$, each mixture component is isotropic $p(\mathbf{x}|\mathbf{h}_n) = \mathcal{N}(\mathbf{x}; \boldsymbol{\mu}_{\mathbf{h}_n}, 0.5^2 I)$ and the cluster identity variables are sampled from a categorial uniform distribution $p(\mathbf{h}_n = j) = \frac{1}{4}$.

We test EP, SEP, SEP with different clipping norms (clipped SEP), and DP-SEP to approximate the joint posterior[1] over the cluster means $\boldsymbol{\mu}_j$ and the cluster identity variables $\mathbf{h}_n$. We also test DP-EM (Park et al., 2017) that adds Gaussian noise to the expected sufficient statistics to ensure the parameters of the mixture of Gaussians model to be differentially private.

Figure 1 visualizes the posterior means (two input dimensions are chosen for visualization) by each of these methods after 100 iterations. For DP-SEP we set the clipping norm to $C = 1$. For SEP and DP-SEP, we

---

[1]Following (Li et al., 2015), we also assume the rest of the parameters to be known.

fixed the damping value, $\gamma = 1$, i.e., $\gamma/N = 1/1000$. The figure shows that for a restrictive privacy regime $\epsilon = 1$, the clusters obtained by DP-SEP are not well separated. However, as we increase the privacy loss, the performance of DP-SEP gets closer to the non-private ones (SEP and EP) and the ground truth. The posterior from DP-SEP exhibits a higher uncertainty than the other non-private methods due to the clipping threshold and the added noise to the mean and covariance during training.

In Table 1, we also provide a quantitative analysis of the results above in terms of F-norm of the difference between the ground truth parameters (Gaussian parameters fitted by No-U-Turn Sampler (NUTS) (Hoffman & Gelman, 2014)) and the estimated parameters by each method. F-norm is first used to evaluate the accuracy of the learned posterior in (Li et al., 2015). It is the L2 distance between the parameters of the ground truth posterior and those of the learned posterior, when the parameters are flattened into a single long vector. The values reported in Table 1 are averages across five independent runs of each method.

As one could expect, as the dataset size is relatively small $N = 1000$ but the number of posterior parameters is relatively large, the privacy-accuracy trade-off measured in terms of F-norm is poor. However, in the next experiment with large datasets, this is not the case.

Table 2: Regression datasets. Size, number of numerical features.

| Dataset | # samps | # features |
|---------|---------|-----------|
| Naval | 11934 | 16 |
| Kin8nm | 8192 | 8 |
| Power | 9568 | 4 |
| Wine | 1599 | 11 |
| Protein | 45730 | 9 |
| Year | 515345 | 90 |

## 6.2 Bayesian neural networks

We explore the performance of DP-SEP on neural network models to handle more complex real-world datasets for regression problems. The datasets used in the experiments are publicly available at the UCI machine learning repository[2] and a brief description can be found in Table 2.

We consider a fully-connected neural network model with 1 hidden layer, which consists of input layer that maps inputs to the hidden units and a hidden layer that maps hidden units' output to the output of the network. Under this neural network model, a mini-batch of the data $X_s \in \mathbb{R}^{s \times d}$ propagates through the network by first going through the input layer, then the hidden layer sequentially given by

$$\text{input layer's output}: Z_0 = \sigma(X_s W_0^\top), \tag{9}$$

$$\text{network's output}: \mathbf{z}_1 = Z_0^\top \mathbf{w}_1^\top, \tag{10}$$

where $\sigma$ is a element-wise non-linearity such as sigmoid or rectifying linear unit (ReLU) and the weight matrix of the input layer is $W_0$ and and weight vector of the hidden layer is $\mathbf{w}_1$. Note that the size of $Z_0$ is the number of hidden units $d_h$ by the input dimension[3] $d$. The size of the network's output is the mini-batch size $s$, as the output of the network given a datapoint is 1-dimensional in the regression problems we consider here. We set the number of hidden units to 100 for *Year* and *Protein* datasets and to 50 for the other four UCI datasets we used. The likelihood of the mini-batch of the data under the neural network model is given by

$$p(\mathbf{y}|W_0, \mathbf{w}_1, X_s, \gamma) = \prod_{i=1}^{s} \mathcal{N}(y_i | z_{1,i}, \gamma^{-1}) \tag{11}$$

where $\gamma$ is the noise precision and $z_{1,i} = \mathbf{w}_1^\top \sigma(W_0 \mathbf{x}_i)$.

---

[2]https://archive.ics.uci.edu/ml/index.php
[3]For simplicity in notation, we do not include the bias term. However, in our code, we use the bias term in each layer.

Table 3: RMSE on test data at $\epsilon = 1$ and $\delta = 1e^{-5}$ (UCI datasets)

| Dataset | Average Test RMSE and Standard deviation | | | | | |
|---|---|---|---|---|---|---|
| | VI | EP | SEP | SEP clipped | DP-SEP | DP-VI |
| Naval | $0.005 \pm 0.0005$ | $0.003 \pm 0.0002$ | $\mathbf{0.002 \pm 0.0001}$ | $0.002 \pm 0.0002$ | $0.002 \pm 0.0003$ | $0.010 \pm 0.0016$ |
| Kin8nm | $0.099 \pm 0.0009$ | $0.088 \pm 0.0044$ | $0.089 \pm 0.0042$ | $0.078 \pm 0.0033$ | $\mathbf{0.078 \pm 0.0022}$ | $0.098 \pm 0.0085$ |
| Power | $4.327 \pm 0.0352$ | $4.098 \pm 0.1388$ | $4.061 \pm 0.1356$ | $\mathbf{4.013 \pm 0.1246}$ | $4.032 \pm 0.1385$ | $4.350 \pm 0.1274$ |
| Wine | $0.646 \pm 0.0081$ | $\mathbf{0.614 \pm 0.0382}$ | $0.623 \pm 0.0436$ | $0.627 \pm 0.0411$ | $0.627 \pm 0.0362$ | $0.734 \pm 0.0510$ |
| Protein | $4.842 \pm 0.0305$ | $4.654 \pm 0.0572$ | $4.602 \pm 0.0649$ | $4.581 \pm 0.0599$ | $4.585 \pm 0.0589$ | $4.934 \pm 0.0532$ |
| Year | $9.034 \pm$ NA | $8.865 \pm$ NA | $8.873 \pm$ NA | $\mathbf{8.862 \pm NA}$ | $\mathbf{8.862 \pm NA}$ | $9.971 \pm$ NA |

Table 4: Test log-likelihood $\epsilon = 1$ and $\delta = 1e^{-5}$ (UCI datasets)

| Dataset | Avgerage Test Log-likelihood and Standard deviation | | | | | |
|---|---|---|---|---|---|---|
| | VI | EP | SEP | SEP clipped | DP-SEP | DP-VI |
| Naval | $3.734 \pm 0.116$ | $4.164 \pm 0.0556$ | $4.609 \pm 0.0531$ | $\mathbf{4.710 \pm 0.0746}$ | $4.686 \pm 0.1053$ | $3.253 \pm 0.1248$ |
| Kin8nm | $0.897 \pm 0.010$ | $1.007 \pm 0.0486$ | $0.999 \pm 0.0479$ | $1.121 \pm 0.0332$ | $\mathbf{1.125 \pm 0.0212}$ | $0.928 \pm 0.0446$ |
| Power | $-2.890 \pm 0.010$ | $-2.830 \pm 0.0313$ | $-2.821 \pm 0.0316$ | $-2.809 \pm 0.0293$ | $\mathbf{-2.814 \pm 0.0323}$ | $-3.077 \pm 0.0816$ |
| Wine | $-0.980 \pm 0.013$ | $\mathbf{-0.926 \pm 0.0487}$ | $-0.936 \pm 0.0643$ | $-0.938 \pm 0.0581$ | $-0.938 \pm 0.0486$ | $-1.213 \pm 0.0831$ |
| Protein | $-2.992 \pm 0.006$ | $-2.957 \pm 0.0121$ | $-2.945 \pm 0.0139$ | $-2.941 \pm 0.0128$ | $\mathbf{-2.941 \pm 0.0130}$ | $-3.049 \pm 0.0382$ |
| Year | $-3.622 \pm$ NA | $-3.604 \pm$ NA | $-3.599 \pm$ NA | $-3.598 \pm$ NA | $\mathbf{-3.597 \pm NA}$ | $-3.974 \pm$ NA |

The first comparison method is *probabilistic backpropagation (PBP)* (Hernández-Lobato & Adams, 2015), an approximate Bayesian inference framework for neural network models. In PBP, each element of the weight matrices in all layers (input layer and hidden layer if we have one-hidden layer) is assumed to be Gaussian distributed with a scalar precision parameter in the prior distribution. Furthermore, a Gamma distribution is imposed on the noise precision parameter as a prior distribution and on the precision parameter as a hyper-prior distribution:

$$p(W_0) = \prod_{i=1}^{h_d} \prod_{j}^{d} \mathcal{N}(w_{0,ij}|0, \lambda^{-1}), \tag{12}$$

$$p(\mathbf{w}_1) = \prod_{k=1}^{h_d} \mathcal{N}(w_{1,k}|0, \lambda^{-1}), \tag{13}$$

$$p(\lambda) = \text{Gam}(\lambda|\alpha_0^\lambda, \beta_0^\lambda), \tag{14}$$

$$p(\gamma) = \text{Gam}(\gamma|\alpha_0^\gamma, \beta_0^\gamma) \tag{15}$$

The approximate posterior is assumed to be factorized for computational tractability, given by

$$q(W_0, \mathbf{w}_1, \lambda, \gamma)$$
$$= \prod_{i=1}^{h_d} \prod_{j}^{d} \mathcal{N}(w_{0,ij}|m_{ij}, v_{ij}) \prod_{k=1}^{h_d} \mathcal{N}(w_{1,k}|m'_k, v'_k)$$
$$\text{Gam}(\lambda|\alpha^\lambda, \beta^\lambda)\text{Gam}(\gamma|\alpha^\gamma, \beta^\gamma), \tag{16}$$

where $\{m_{ij}, v_{ij}, m'_k, v'_k, \alpha^\lambda, \beta^\lambda, \alpha^\gamma, \alpha^\gamma\}$ are posterior parameters.

PBP uses *assumed density filtering* (ADF) for estimating the posterior parameters (Maybeck, 1982). ADF can be viewed as a simpler version of EP. It maintains a global approximation only and as a result performs poorer than EP.

We use the same prior and posterior configurations as in PBP in other comparison method such as EP, SEP, SEP with only clipping, a scalable VI method for neural networks described in (Graves, 2011) and it's privatized version, DP-VI. We derive the variational inference procedure in Sec. D in Appendix and implement the differentially private version of it in PyTorch.

We consider the 90% of the original dataset randomly subsampled without replacement as a training dataset and the remaining 10% as a test dataset. All the training datasets are normalized to have zero mean and unit variance on their input features and targets.

For SEP, clipped SEP and DP-SEP, we fix the damping factor to $1/N$. We also fix the clipping norm to $C = 1$ for DP-SEP. Further experiments on DP-SEP with different clipping norms can be found in Sec. E in Appendix. We set the privacy budget to $\epsilon = 1$ and $\delta = 10^{-5}$ in DP-SEP. In DP-VI experiments we fixed $\delta = 10^{-5}$ and set $\sigma$ value to get a final $\epsilon \approx 1$. The detailed hyper-parameter setting for VI and DP-VI experiments can be found in Table 6 and Table 7 in Sec. D.1 in Appendix.

Table 3 and Table 4 shows the average test RMSE and test log-likelihood after 10 independent runs for each dataset except for *Year*, where only one split is performed according to the recommendations of the dataset[4].

When comparing between non-private algorithms, the RMSE and test log-likelihood indicates that EP based methods (EP, SEP and SEP clipped) give better posterior estimates than those obtained by VI. In terms of DP-SEP performance over the different datasets, the results show that our algorithm gives better approximates than those from DP-VI and that it is comparable to SEP and even better in some cases as for *Kin8nm*. In fact, clipping the norm of the natural parameters and the intermediate approximating factor on the SEP algorithm has a positive effect on the original algorithm and reduces the test averaged RMSE in most cases. This seems to indicate that clipping acts as a regularizer (or a constraint) for the posterior to be well concentrated.

In Table 5 in Sec. C in Appendix, we also show the point estimate (through the usual stochastic gradient descent) which helps gauging how well these approximate inference methods are performing in an absolute sense. The results show that in most of the UCI datasets we considered, DP-SEP outperforms the point estimate.

## 7 Conclusions and future work

In this work, we have presented differentially private stochastic expectation propagation (DP-SEP), a novel algorithm to perform private approximate Bayesian inference based on the SEP algorithm. In DP-SEP, we add carefully calibrated noise to natural parameters to obtain a differentially private posterior distribution. We provide a theoretical analysis on how the noise added for privacy affects the accuracy on the posterior distribution.

The clustering experiments under the Mixture of Gaussians model with a relatively small synthetic dataset shows that DP-SEP produces approximate posterior distributions that present higher uncertainty than those generated by non-private methods due to the poor sensitivity. We also provide quantitative results comparing the ground truth parameters and the posterior parameters by DP-SEP, where relaxing the privacy constraints improves the accuracy of the private posterior distributions. We also test DP-SEP on real-world datasets for regression tasks by implementing DP-SEP for a neural network model. The results show that DP-SEP often yields the posterior distributions that are better than those by SEP, thanks to the help of clipping natural parameters and large dataset sizes.

SEP provides better uncertainty estimates than variational inference, which was demonstrated in the experiments we presented as well as in existing literature. We look forward to extending our private SEP to more large-scale scenarios such as federated learning settings, which can be done in a straightforward way.

---

[4]See: https://archive.ics.uci.edu/ml/datasets/yearpredictionmsd

**Acknowledgments**

We thank the support, computational resources, and services provided by the Canada CIFAR AI Chairs program (at AMII) and the Digital Research Alliance of Canada (Compute Canada). This work was done while M. Vinaroz was a visiting international research student in the department of computer science at University of British Columbia. We thank our anonymous reviewers for their constructive feedback. Their feedback helped us improve our manuscript significantly.

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

# Appendix

## A    KL between SEP and clipped SEP

Denote the posterior distribution of SEP by $q := \mathcal{N}(\boldsymbol{\mu}_q, \Sigma_q)$ and the posterior distribution of SEP with clipping threshold $C$ by $p := \mathcal{N}(\boldsymbol{\mu}_p, \Sigma_p)$ where the following relation for natural parameters holds: $\eta_p = \eta_q / \max\left(1, \frac{\|\eta_q\|_2}{C}\right)$ and $\Lambda_p = \Lambda_q / \max\left(1, \frac{\|\Lambda_q\|_F}{C}\right)$. Then the KL-divergence between $q$ and $p$ can be expressed in terms of $q$ by:

$$D_{kl}[q||p] = \frac{1}{2}\left[d\max\left(1, \frac{\|\Lambda_q\|_F}{C}\right) + b^2/\max\left(1, \frac{\|\Lambda_q\|_F}{C}\right)(\Lambda_q^{-1}\eta_q)^\top\eta_q - d + d\log\max\left(1, \frac{\|\Sigma_q\|_F}{C}\right)\right]$$

where $b = \max\left(1, \frac{\|\Lambda_q\|_F}{C}\right)/\max\left(1, \frac{\|\eta_q\|_F}{C}\right) - 1$

*Proof.* The closed form for the KL-divergence is:

$$D_{kl}[q||p] = \frac{1}{2}\left[Tr(\Sigma_p^{-1}\Sigma_q) + (\mu_p - \mu_q)^\top\Sigma_p^{-1}(\mu_p - \mu_q) - d + \log\left(\frac{|\Sigma_p|}{|\Sigma_q|}\right)\right]$$

we can rewrite the KL-divergence in terms of the natural parameters ($\Sigma = \Lambda^{-1}$ and $\mu = \Lambda^{-1}\eta$) :

$$D_{kl}[q||p] = \frac{1}{2}\left[Tr(\Lambda_p\Lambda_q^{-1}) + (\Lambda_p^{-1}\eta_p - \Lambda_q^{-1}\eta_q)^\top\Lambda_p(\Lambda_p^{-1}\eta_p - \Lambda_q^{-1}\eta_q) - d + \log\left(\frac{|\Lambda_p^{-1}|}{|\Lambda_q^{-1}|}\right)\right] \qquad (17)$$

We have by definition of $\Lambda_p$ that $\Lambda_p^{-1} = \left[\Lambda_q/\max\left(1, \frac{\|\Lambda_q\|_F}{C}\right)\right]^{-1} = \max\left(1, \frac{\|\Lambda_q\|_F}{C}\right)\Lambda_q^{-1}$ and

$|\Lambda_p^{-1}| = \left[\max\left(1, \frac{\|\Sigma_q\|_F}{C}\right)\right]^d |\Lambda_q^{-1}|$ so the logarithmic term becomes:

$$\log\left(\frac{|\Lambda_p^{-1}|}{|\Lambda_q^{-1}|}\right) = \log(\max\left(1, \frac{\|\Sigma_q\|_F}{C}\right)^d) = d\log\max\left(1, \frac{\|\Sigma_q\|_F}{C}\right)$$

For the trace term we have:
$$Tr(\Lambda_p\Lambda_q^{-1}) = Tr(1/\max\left(1, \frac{\|\Lambda_q\|_F}{C}\right)\Lambda_q\Lambda_q^{-1}) = Tr(1/\max\left(1, \frac{\|\Lambda_q\|_F}{C}\right)I_d) = d\max\left(1, \frac{\|\Lambda_q\|_F}{C}\right)$$

$$(\Lambda_p^{-1}\eta_p - \Lambda_q^{-1}\eta_q) = \left[\max\left(1, \frac{\|\Lambda_q\|_F}{C}\right)/\max\left(1, \frac{\|\eta_q\|_F}{C}\right) - 1\right]\Lambda_q^{-1}\eta_q = b\Lambda_q^{-1}\eta_q$$

Thus the quadratic term can be rewritten as:

$$(\Lambda_p^{-1}\eta_p - \Lambda_q^{-1}\eta_q)^\top\Lambda_p(\Lambda_p^{-1}\eta_p - \Lambda_q^{-1}\eta_q) = b^2/\max\left(1, \frac{\|\Lambda_q\|_F}{C}\right)(\Lambda_q^{-1}\eta_q)^\top\Lambda_q\Lambda_q^{-1}\eta_q$$

$$= b^2/\max\left(1, \frac{\|\Lambda_q\|_F}{C}\right)(\Lambda_q^{-1}\eta_q)^\top\eta_q$$

Then the KL-divergence can be expressed in terms of the natural parameters of q by:

$$D_{kl}[q||p] = \frac{1}{2}\left[d\max\left(1, \frac{\|\Lambda_q\|_F}{C}\right) + b^2/\max\left(1, \frac{\|\Lambda_q\|_F}{C}\right)(\Lambda_q^{-1}\eta_q)^T\eta_q - d + d\log\max\left(1, \frac{\|\Sigma_q\|_F}{C}\right)\right]$$

$\square$

## B  Proof of Thm. 4.1

*Proof.* Denote the posterior distribution from SEP by $q := \mathcal{N}(\boldsymbol{\mu}_q, \Sigma_q)$ and the posterior distribution of DP-SEP to be $p := \mathcal{N}(\boldsymbol{\mu}_p, \Sigma_p)$. Mean and moment parameters of $q$ can be expressed in terms of the natural parameters by $\Sigma_q = \Lambda_q^{-1}$ and $\mu_q = \Lambda_q^{-1}\eta_q$ and the those of $p$ verify the following relations:

$$\eta_p = \eta_q + \mathbf{e} \text{ where } \mathbf{e}_i \sim \mathcal{N}(0, \sigma_1^2)$$

$$\Lambda_p = \Lambda_q + \mathbf{E} + A \text{ where } \mathbf{E}_{ij} \sim \mathcal{N}(0, \sigma_2^2)$$

Here, we define

$$A = (-\lambda_{min}(\Lambda_q + \mathbf{E}) + \rho)I \tag{18}$$

to ensure that $\Lambda_p$ is positive definite by taking the minimum eigenvalue of $\Lambda_q + \mathbf{E}$ and adding a small positive constant $\rho$ such that $\lambda_{min}(\Lambda_q + \mathbf{E} + A) \geq \rho$.

The KL-divergence is written in closed form in terms of natural parameters:

$$\mathbb{E}_{\mathbf{e}}\mathbb{E}_{\mathbf{E}}(D_{kl}[q||p])$$
$$= \frac{1}{2}\left[\mathbb{E}_{\mathbf{E}}(Tr(\Lambda_p\Lambda_q^{-1})) + \mathbb{E}_{\mathbf{e}}\mathbb{E}_{\mathbf{E}}((\Lambda_p^{-1}\eta_p - \Lambda_q^{-1}\eta_q)^\top\Lambda_p(\Lambda_p^{-1}\eta_p - \Lambda_q^{-1}\eta_q))\right.$$
$$\left. + \mathbb{E}_{\mathbf{E}}\left(\log\left(\frac{|\Lambda_q|}{|\Lambda_p|}\right)\right) - d\right]. \tag{19}$$

In bounding some of these terms, we rely on the tail probability of a Gaussian random matrix. First, we re-formulate $\mathbf{E}$ as a matrix Gaussian series where

$$\mathbf{E} := \sum_{k=1}^{\frac{d(d+1)}{2}} \sigma_2\gamma_k B_k, \tag{20}$$

where $\gamma_k$ is a standard normal Gaussian random variable and $\{B_k\}$ is a finite sequence of fixed symmetric (Hermitian) matrices. Then, due to Theorem 4.6.1. in Tropp (2015),

$$\mathbb{E}_{\mathbf{E}}[\lambda_{max}(\mathbf{E})] \leq \sqrt{2v(\mathbf{E})\log(d)}, \tag{21}$$

where the matrix variance statistic of the sum is denoted by $v(\mathbf{E}) = \|\sum_k \sigma_2^2 B_k B_k^T\|$ and by symmetry (since $-\mathbf{E}$ has the same distribution as $\mathbf{E}$),

$$\mathbb{E}_{\mathbf{E}}[\lambda_{min}(\mathbf{E})] = \mathbf{E}_{\mathbf{E}}[\lambda_{min}(-\mathbf{E})],$$
$$= -\mathbb{E}_{\mathbf{E}}[\lambda_{max}(\mathbf{E})],$$
$$\geq -\sqrt{2v(\mathbf{E})\log(d)}. \tag{22}$$

Furthermore, for all $M \geq 0$:

$$P(\lambda_{max}(\mathbf{E}) \leq M) \geq 1 - d\exp\left(\frac{-M^2}{2v(\mathbf{E})}\right) \tag{23}$$

and

$$P(\lambda_{min}(\mathbf{E}) \geq -M) \geq 1 - d \exp\left(\frac{-M^2}{2v(\mathbf{E})}\right) \tag{24}$$

Now we take a look at each term below.

**First term:**

$$\mathbb{E}_{\mathbf{E}}(Tr(\Lambda_p \Lambda_q^{-1})) \tag{25}$$

$$= \mathbb{E}_{\mathbf{E}} Tr(I + \mathbf{E}\Lambda_q^{-1} + A\Lambda_q^{-1}) \tag{26}$$

$$= Tr(I) + \mathbb{E}_{\mathbf{E}}(Tr(\mathbf{E}\Lambda_q^{-1})) + \mathbb{E}_{\mathbf{E}} Tr(A\Lambda_q^{-1}) \tag{27}$$

$$= Tr(I) + Tr(\mathbb{E}_{\mathbf{E}}(\mathbf{E}\Lambda_q^{-1})) + Tr(\mathbb{E}_{\mathbf{E}}(A\Lambda_q^{-1})), \tag{28}$$

due to `https://statproofbook.github.io/P/mean-tr` $\tag{29}$

$$= d + Tr(\mathbb{E}_{\mathbf{E}}((-\lambda_{min}(\Lambda_q + \mathbf{E}) + \rho)I\Lambda_q^{-1})), \text{ since } \mathbb{E}_{\mathbf{E}}(\mathbf{E}\Lambda_q^{-1}) = 0, \text{due to eq. 18} \tag{30}$$

$$\leq d + Tr(\mathbb{E}_{\mathbf{E}}((-\lambda_{min}(\Lambda_q) - \lambda_{min}(\mathbf{E}) + \rho))\Lambda_q^{-1}), \text{ by Weyl's inequality} \tag{31}$$

$$= d + Tr((-\lambda_{min}(\Lambda_q) - \mathbb{E}_{\mathbf{E}}(\lambda_{min}(\mathbf{E})) + \rho)\Lambda_q^{-1}) \tag{32}$$

$$\leq d + Tr((-\lambda_{min}(\Lambda_q) + \sqrt{2v(\mathbf{E})\log(d)} + \rho)\Lambda_q^{-1}), \text{ due to eq. 22} \tag{33}$$

**Second term:**

$$\mathbb{E}_{\mathbf{e}}\mathbb{E}_{\mathbf{E}}((\Lambda_p^{-1}\eta_p - \Lambda_q^{-1}\eta_q)^\top \Lambda_p(\Lambda_p^{-1}\eta_p - \Lambda_q^{-1}\eta_q))$$

$$= \mathbb{E}_{\mathbf{e}}\mathbb{E}_{\mathbf{E}}((\eta_p^\top \Lambda_p^{-1} - \eta_q^\top \Lambda_q^{-1})(\eta_p - \Lambda_p\Lambda_q^{-1}\eta_q))$$

$$= \mathbb{E}_{\mathbf{e}}\mathbb{E}_{\mathbf{E}}(((\eta_q + \mathbf{e})^\top \Lambda_p^{-1} - \eta_q^\top \Lambda_q^{-1})((\eta_q + \mathbf{e}) - \Lambda_p\Lambda_q^{-1}\eta_q))$$

$$= \mathbb{E}_{\mathbf{e}}\mathbb{E}_{\mathbf{E}}((\eta_q^\top \Lambda_p^{-1} + \mathbf{e}^\top \Lambda_p^{-1} - \eta_q^\top \Lambda_q^{-1})(\eta_q + \mathbf{e} - \Lambda_p\Lambda_q^{-1}\eta_q))$$

$$= \mathbb{E}_{\mathbf{e}}\mathbb{E}_{\mathbf{E}}(\eta_q^\top \Lambda_p^{-1}\eta_q + \eta_q^\top \Lambda_p^{-1}\mathbf{e} - \eta_q^\top \Lambda_q^{-1}\eta_q + \mathbf{e}^\top \Lambda_p^{-1}\eta_q + \mathbf{e}^\top \Lambda_p^{-1}\mathbf{e} - \mathbf{e}^\top \Lambda_q^{-1}\eta_q$$

$$- \eta_q^\top \Lambda_q^{-1}\eta_q - \eta_q^\top \Lambda_q^{-1}\mathbf{e} + \eta_q^\top \Lambda_q^{-1}\Lambda_p\Lambda_q^{-1}\eta_q)$$

$$= \mathbb{E}_{\mathbf{E}}(\eta_q^\top \Lambda_p^{-1}\eta_q - \eta_q^\top \Lambda_q^{-1}\eta_q + \sigma_1^2 Tr(\Lambda_p^{-1}) - \eta_q^\top \Lambda_q^{-1}\eta_q + \eta_q^\top \Lambda_q^{-1}\Lambda_p\Lambda_q^{-1}\eta_q)$$

$$= \mathbb{E}_{\mathbf{E}}(\eta_q^\top \Lambda_p^{-1}\eta_q) - \eta_q^\top \Lambda_q^{-1}\eta_q + \sigma_1^2 \mathbb{E}_{\mathbf{E}}(Tr(\Lambda_p^{-1})) - \eta_q^\top \Lambda_q^{-1}\eta_q + \mathbb{E}_{\mathbf{E}}(\eta_q^\top \Lambda_q^{-1}\Lambda_p\Lambda_q^{-1}\eta_q)$$

$$= \mathbb{E}_{\mathbf{E}}(\eta_q^\top \Lambda_p^{-1}\eta_q) - \eta_q^\top \Lambda_q^{-1}\eta_q + \sigma_1^2 \mathbb{E}_{\mathbf{E}}(Tr(\Lambda_p^{-1})) + \mathbb{E}_{\mathbf{E}}(\eta_q^\top \Lambda_q^{-1}A\Lambda_q^{-1}\eta_q)$$

$$\leq \sum_{i=1}^{d} b_i^2 \left(\frac{1 + \frac{\rho - \lambda_{min}(\Lambda_q) + \sqrt{2v(\mathbf{E})\log(d)}}{\lambda_i(\Lambda_q)}}{\left(1 + \frac{-2M - \lambda_{min}(\Lambda_q) + \rho}{\lambda_i(\Lambda_q)}\right)\left(1 + \frac{2M - \lambda_{min}(\Lambda_q) + \rho}{\lambda_i(\Lambda_q)}\right)}\right) - \eta_q^\top \Lambda_q^{-1}\eta_q$$

$$+ \sigma_1^2 \lambda_{max}(\Sigma_q) \sum_{i=1}^{d} \left(\frac{1 + \frac{\rho - \lambda_{min}(\Lambda_q) + \sqrt{2v(\mathbf{E})\log(d)}}{\lambda_i(\Lambda_q)}}{\left(1 + \frac{-2M - \lambda_{min}(\Lambda_q) + \rho}{\lambda_i(\Lambda_q)}\right)\left(1 + \frac{2M - \lambda_{min}(\Lambda_q) + \rho}{\lambda_i(\Lambda_q)}\right)}\right)$$

$$+ \left(\sum_{i=1}^{d} a_i^2\right)\left(-\lambda_{min}(\Lambda_q) + \sqrt{2v(\mathbf{E})\log(d)} + \rho\right) \tag{34}$$

The inequality in the last step is due to:

1. For bounding $\mathbb{E}_{\mathbf{E}}(\eta_q^\top \Lambda_p^{-1}\eta_q)$, we first re-write $\Lambda_p = \Lambda_q^{\frac{1}{2}}\left[I + \Lambda_q^{-\frac{1}{2}}(\mathbf{E} + A)\Lambda_q^{-\frac{1}{2}}\right]\Lambda_q^{\frac{1}{2}}$ and denote

$$\mathbf{b} := \Lambda_q^{-\frac{1}{2}}\eta_q \tag{35}$$

$$P(\mathbf{E}) := \Lambda_q^{-\frac{1}{2}}(\mathbf{E} + A)\Lambda_q^{-\frac{1}{2}} \tag{36}$$

to simplify the notation in the following steps. Thus, we have:

$$\mathbb{E}_{\mathbf{E}}(\mathbf{b}^\top [I + P(\mathbf{E})]^{-1} \mathbf{b}) \tag{37}$$

$$= \mathbb{E}_{\mathbf{E}} \left[ Tr([I + P(\mathbf{E})]^{-1} \mathbf{b}\mathbf{b}^\top) \right] \tag{38}$$

$$= \mathbb{E}_{\mathbf{E}} \left( \sum_i^d \frac{b_i^2}{1 + \lambda_i(P(\mathbf{E}))} \right), \tag{39}$$

$$\leq \left( \sum_i^d b_i^2 \mathbb{E}_{\mathbf{E}} \left[ \frac{1}{1 + \lambda_i(P(\mathbf{E}))} \right] \right), \tag{40}$$

where the last line is because the sum and average are linear operation, we can swap the order. Here $\lambda_i(P(\mathbf{E}))$ denotes the $i$-th eigenvalue of $P(\mathbf{E})$.

Now, we are interested in upper bounding $\mathbb{E}_{\mathbf{E}} \left[ \frac{1}{1+\lambda_i(P(\mathbf{E}))} \right]$. To do so, we use the following (mat, 2019): if a random variable $X$ is bounded by $h \leq X \leq H$, then

$$\mathbb{E} \left[ \frac{1}{X} \right] \leq \frac{H + h - \mathbb{E}(X)}{Hh} \tag{41}$$

Because we are adding Gaussian noise and the domain of Gaussian noise is unbounded, $\lambda_i(P(\mathbf{E}))$ is not strictly bounded. Hence, using the tail bound of random Gaussian matrix, we achieve the bounds with high probability.

Here, we set $X = 1 + \lambda_i(P(\mathbf{E}))$. Recall $\lambda_i(P(\mathbf{E})) = \frac{\lambda_i(\mathbf{E} + A)}{\lambda_i(\Lambda_q)} = \frac{\lambda_i(\mathbf{E}) + \lambda_i(A)}{\lambda_i(\Lambda_q)}$, $i \in \{1, \ldots, d\}$, because $A$ is a diagonal matrix. We need to identify what $h$ and $H$ are that satisfy $h \leq 1 + \lambda_i(P(\mathbf{E})) \leq H$.

First, we know that the following holds:

$$1 + \lambda_i(P(\mathbf{E})) \leq 1 + \frac{\lambda_{max}(\mathbf{E}) + \lambda_{max}(A)}{\lambda_i(\Lambda_q)} \leq 1 + \frac{M + \lambda_{max}(A)}{\lambda_i(\Lambda_q)} \tag{42}$$

where the last inequality is due to eq. 23. Now, we find the upper bound for $\lambda_{max}(A)$:

$$\lambda_{max}(A) \tag{43}$$

$$= \lambda_{max}[(-\lambda_{min}(\Lambda_q + \mathbf{E}) + \rho)I], \text{ by defition of } A \text{ given in eq. 18} \tag{44}$$

$$= -\lambda_{min}(\Lambda_q + \mathbf{E}) + \rho \tag{45}$$

$$\leq -\lambda_{min}(\Lambda_q) - \lambda_{min}(\mathbf{E}) + \rho, \text{ by Weyl's inequality} \tag{46}$$

$$\leq -\lambda_{min}(\Lambda_q) + \lambda_{max}(\mathbf{E}) + \rho, \ -\mathbf{E} \text{ has the same distribution as } \mathbf{E} \tag{47}$$

$$\leq -\lambda_{min}(\Lambda_q) + M + \rho \text{ due to eq. 23} \tag{48}$$

Hence, we establish the upper bound given by

$$1 + \lambda_i(P(\mathbf{E})) \leq H := 1 + \frac{2M + \rho - \lambda_{min}(\Lambda_q)}{\lambda_i(\Lambda_q)} \tag{49}$$

with probability given in eq. 23.

Now, it remains to lower bound, $1 + \lambda_i(P(\mathbf{E}))$. First, we know that $\lambda_i(P(\mathbf{E})) = \lambda_i\left(\dfrac{\mathbf{E} + A}{\Lambda_q}\right) = \dfrac{\lambda_i(\mathbf{E}) + \lambda_i(A)}{\lambda_i(\Lambda_q)} \geq \dfrac{\lambda_{min}(\mathbf{E}) + \lambda_{min}(A)}{\lambda_i(\Lambda_q)}$. Let's find a lower bound for $\lambda_{min}(\mathbf{E}) + \lambda_{min}(A)$. As before,

$$\lambda_{min}(A) \tag{50}$$
$$= \lambda_{min}[(-\lambda_{min}(\Lambda_q + \mathbf{E}) + \rho)I], \tag{51}$$
$$= -\lambda_{min}(\Lambda_q + \mathbf{E}) + \rho \tag{52}$$
$$\geq -\lambda_{min}(\Lambda_q) - \lambda_{max}(\mathbf{E}) + \rho), \text{ by Weyl's inequality} \tag{53}$$
$$\geq -\lambda_{min}(\Lambda_q) + \lambda_{min}(\mathbf{E}) + \rho, \text{ since } -\mathbf{E} \text{ has the same distribution as } \mathbf{E} \tag{54}$$
$$\geq -\lambda_{min}(\Lambda_q) - M + \rho, \text{ due to eq. 24} \tag{55}$$

From eq. 24 we have that $\lambda_{min}(\mathbf{E}) \geq -M$ with probability at least $1 - d\exp\left(\frac{-M^2}{2v(\mathbf{E})}\right)$. Hence, we arrive at:

$$\lambda_{min}(\mathbf{E} + A) = \lambda_{min}(\mathbf{E}) + \lambda_{min}(A) \geq -\lambda_{min}(\Lambda_q) - 2M + \rho, \tag{56}$$

Assuming $-\lambda_{min}(\Lambda_q) - 2M + \rho \leq 0$ (as we set $\rho$ to be small),

$$\lambda_i(P(\mathbf{E})) \geq \frac{\lambda_{min}(\mathbf{E}) + \lambda_{min}(A)}{\lambda_i(\Lambda_q)} \geq \frac{-2M + \rho - \lambda_{min}(\Lambda_q)}{\lambda_i(\Lambda_q)}. \tag{57}$$

Hence, the lower bound is

$$h := 1 + \frac{-2M + \rho - \lambda_{min}(\Lambda_q)}{\lambda_i(\Lambda_q)} \leq 1 + \lambda_{min}(P(\mathbf{E})) \leq 1 + \lambda_i(P(\mathbf{E})), \tag{58}$$

with probability given in eq. 24.

Hence, using the bound in eq. 41, we bound the following:

$$\mathbb{E}_{\mathbf{E}}\left(\frac{1}{1 + \lambda_i(P(\mathbf{E}))}\right) \tag{59}$$

$$\leq \frac{1 + \frac{-2M - \lambda_{min}(\Lambda_q) + \rho}{\lambda_i(\Lambda_q)} + 1 + \frac{2M - \lambda_{min}(\Lambda_q) + \rho}{\lambda_i(\Lambda_q)} - 1 - \mathbb{E}_{\mathbf{E}}\left(\lambda_i(P(\mathbf{E}))\right)}{\left(1 + \frac{-2M - \lambda_{min}(\Lambda_q) + \rho}{\lambda_i(\Lambda_q)}\right)\left(1 + \frac{2M - \lambda_{min}(\Lambda_q) + \rho}{\lambda_i(\Lambda_q)}\right)} \tag{60}$$

$$= \frac{1 + \frac{2\rho - 2\lambda_{min}(\Lambda_q)}{\lambda_i(\Lambda_q)} - \mathbb{E}_{\mathbf{E}}\left(\lambda_i(P(\mathbf{E}))\right)}{\left(1 + \frac{-2M - \lambda_{min}(\Lambda_q) + \rho}{\lambda_i(\Lambda_q)}\right)\left(1 + \frac{2M - \lambda_{min}(\Lambda_q) + \rho}{\lambda_i(\Lambda_q)}\right)} \tag{61}$$

with probability at least $1 - d\exp\left(\frac{-M^2}{2v(\mathbf{E})}\right)$. Now, it remains to upper bound the expectation term in $-\mathbb{E}_E[\lambda_i(P(\mathbf{E}))]$. Following the same trick as before, we lower bound $\mathbb{E}_{\mathbf{E}}[\lambda_i(P(\mathbf{E}))]$:

$$\mathbb{E}_{\mathbf{E}}[\lambda_i(P(\mathbf{E}))] \tag{62}$$
$$= \frac{1}{\lambda_i(\Lambda_q)}\mathbb{E}_{\mathbf{E}}\lambda_i(A + \mathbf{E}), \text{ by definition of } P(\mathbf{E}) \tag{63}$$
$$= \frac{1}{\lambda_i(\Lambda_q)}(\mathbb{E}_{\mathbf{E}}\lambda_i(A) + \mathbb{E}_{\mathbf{E}}\lambda_i(\mathbf{E})), \tag{64}$$
$$\geq \frac{1}{\lambda_i(\Lambda_q)}(\mathbb{E}_{\mathbf{E}}\lambda_i(A) + \mathbb{E}_{\mathbf{E}}\lambda_{min}(\mathbf{E})), \text{ because } \mathbb{E}_{\mathbf{E}}\lambda_{min}(\mathbf{E}) \leq \mathbb{E}_{\mathbf{E}}\lambda_i(\mathbf{E}) \leq \mathbb{E}_{\mathbf{E}}\lambda_{max}(\mathbf{E}) \tag{65}$$
$$\geq \frac{1}{\lambda_i(\Lambda_q)}(\mathbb{E}_{\mathbf{E}}\lambda_i(A) - \sqrt{2v(\mathbf{E})\log(d)}), \text{ because of eq. 22} \tag{66}$$
$$\geq \frac{1}{\lambda_i(\Lambda_q)}(-\lambda_{min}(\Lambda_q) - \sqrt{2v(\mathbf{E})\log(d)} + \rho), \tag{67}$$

where

$$\mathbb{E}_{\mathbf{E}}[\lambda_i(A)] \tag{68}$$

$$= \mathbb{E}_{\mathbf{E}}[-\lambda_{min}(\Lambda_q + \mathbf{E}) + \rho], \text{ by definition of } A \tag{69}$$

$$\geq -\lambda_{min}(\mathbb{E}_{\mathbf{E}}(\Lambda_q + \mathbf{E})) + \rho, \tag{70}$$

$$\text{since } \lambda_{min} \text{ is a concave function and } -\lambda_{min} \text{ is convex. With Jensen's inequality} \tag{71}$$

$$\geq -\lambda_{min}(\Lambda_q) + \rho, \text{ since } \mathbb{E}_{\mathbf{E}}(\mathbf{E}) = 0. \tag{72}$$

2. As before, we re-write $\Lambda_p = \Lambda_q^{\frac{1}{2}} \left[ I + \Lambda_q^{-\frac{1}{2}} (\mathbf{E} + A)\Lambda_q^{-\frac{1}{2}} \right] \Lambda_q^{\frac{1}{2}}$. The inverse is $\Lambda_p^{-1} = \Lambda_q^{-\frac{1}{2}} \left[ I + \Lambda_q^{-\frac{1}{2}} (\mathbf{E} + A)\Lambda_q^{-\frac{1}{2}} \right]^{-1} \Lambda_q^{-\frac{1}{2}}$. Therefore,

$$\mathbb{E}_{\mathbf{E}}(Tr(\Lambda_p^{-1}))$$

$$= \mathbb{E}_{\mathbf{E}}(Tr(\Lambda_q^{-\frac{1}{2}} \left[ I + \Lambda_q^{-\frac{1}{2}} (E + A)\Lambda_q^{-\frac{1}{2}} \right]^{-1} \Lambda_q^{-\frac{1}{2}})) \tag{73}$$

$$= \mathbb{E}_{\mathbf{E}}(Tr(\left[ I + \Lambda_q^{-\frac{1}{2}} (\mathbf{E} + A)\Lambda_q^{-\frac{1}{2}} \right]^{-1} \Sigma_q)), \text{ due to the Cyclic property of Trace and } \Sigma_q = \Lambda_q^{-1} \tag{74}$$

$$\leq \mathbb{E}_{\mathbf{E}} \left( \sum_{i=1}^{d} \frac{\lambda_i(\Sigma_q)}{1 + \lambda_i(P(\mathbf{E}))} \right), \text{ due to eq. 35} \tag{75}$$

$$\leq \lambda_{max}(\Sigma_q) \sum_{i=1}^{d} \left( \mathbb{E}_{\mathbf{E}} \left[ \frac{1}{1 + \lambda_i(P(\mathbf{E}))} \right] \right) \tag{76}$$

$$\leq \lambda_{max}(\Sigma_q) \sum_{i=1}^{d} \left( \frac{1 + \frac{2\rho - 2\lambda_{min}(\Lambda_q)}{\lambda_i(\Lambda_q)} + \frac{\sqrt{2v(\mathbf{E})\log(d)} + \lambda_{min}(\Lambda_q) - \rho}{\lambda_i(\Lambda_q)}}{\left( 1 + \frac{-2M - \lambda_{min}(\Lambda_q) + \rho}{\lambda_i(\Lambda_q)} \right) \left( 1 + \frac{2M - \lambda_{min}(\Lambda_q) + \rho}{\lambda_i(\Lambda_q)} \right)} \right), \text{ due to eq. 59} \tag{77}$$

$$\leq \lambda_{max}(\Sigma_q) \sum_{i=1}^{d} \left( \frac{1 + \frac{\rho - \lambda_{min}(\Lambda_q) + \sqrt{2v(\mathbf{E})\log(d)}}{\lambda_i(\Lambda_q)}}{\left( 1 + \frac{-2M - \lambda_{min}(\Lambda_q) + \rho}{\lambda_i(\Lambda_q)} \right) \left( 1 + \frac{2M - \lambda_{min}(\Lambda_q) + \rho}{\lambda_i(\Lambda_q)} \right)} \right), \tag{78}$$

3.

$$\mathbb{E}_{\mathbf{E}}(\eta_q^\top \Lambda_q^{-1} A \Lambda_q^{-1} \eta_q) \tag{79}$$

$$= \mathbb{E}_{\mathbf{E}}(\mathbf{a}^\top A \mathbf{a}), \text{ where } \mathbf{a} = \Lambda_q^{-1} \eta_q \tag{80}$$

$$= \mathbb{E}_{\mathbf{E}} \sum_{i=1}^{d} a_i^2 (-\lambda_{min}(\Lambda_q + \mathbf{E}) + \rho), \text{ due to the definition of } A \tag{81}$$

$$= \left( \sum_{i=1}^{d} a_i^2 \right) \mathbb{E}_{\mathbf{E}}(-\lambda_{min}(\Lambda_q + \mathbf{E}) + \rho), \text{ since expectation is a linear operation and } \mathbf{a} \text{ is constant in } \mathbf{E} \tag{82}$$

$$\leq \left( \sum_{i=1}^{d} a_i^2 \right) (-\lambda_{min}(\Lambda_q) - \mathbb{E}_{\mathbf{E}}[\lambda_{min}(\mathbf{E})] + \rho), \tag{83}$$

by Weyl's inequality and since expectation preserves inequality \tag{84}

$$= \left( \sum_{i=1}^{d} a_i^2 \right) (-\lambda_{min}(\Lambda_q) + \mathbb{E}_{\mathbf{E}}[\lambda_{max}(\mathbf{E})] + \rho), \text{ due to the symmetry of } \mathbf{E} \tag{85}$$

$$\leq \left( \sum_{i=1}^{d} a_i^2 \right) \left( -\lambda_{min}(\Lambda_q) + \sqrt{2v(\mathbf{E})\log(d)} + \rho \right), \text{ due to eq. 21} \tag{86}$$

**Third term:**

Since $\Lambda_p = \Lambda_q^{\frac{1}{2}} \left[ I + \Lambda_q^{-\frac{1}{2}} (\mathbf{E} + A) \Lambda_q^{-\frac{1}{2}} \right] \Lambda_q^{\frac{1}{2}}$,

$$\mathbb{E}_{\mathbf{E}} \left( \log \left( \frac{|\Lambda_q|}{|\Lambda_p|} \right) \right) \tag{87}$$

$$= \mathbb{E}_{\mathbf{E}} \left( \log \left( \frac{|\Lambda_q|}{|\Lambda_q^{\frac{1}{2}} \left[ I + \Lambda_q^{-\frac{1}{2}} (\mathbf{E} + A) \Lambda_q^{-\frac{1}{2}} \right] \Lambda_q^{\frac{1}{2}}|} \right) \right) \tag{88}$$

$$= \mathbb{E}_{\mathbf{E}} \sum_{i=1}^{d} \log \left( \frac{1}{1 + \lambda_i(P(\mathbf{E}))} \right), \text{ due to eq. 35} \tag{89}$$

$$= \sum_{i=1}^{d} \mathbb{E}_{\mathbf{E}} \log \left( \frac{1}{1 + \lambda_i(P(\mathbf{E}))} \right), \text{ swapping } \mathbb{E} \text{ and } \sum_i \tag{90}$$

$$\leq \sum_{i=1}^{d} \log \left( \mathbb{E}_{\mathbf{E}} \left[ \frac{1}{1 + \lambda_i(P(\mathbf{E}))} \right] \right), \text{ because log is concave} \tag{91}$$

$$\leq \sum_{i=1}^{d} \log \left( \frac{1 + \frac{\rho - \lambda_{min}(\Lambda_q) + \sqrt{2v(\mathbf{E})\log(d)}}{\lambda_i(\Lambda_q)}}{\left( 1 + \frac{-2M - \lambda_{min}(\Lambda_q) + \rho}{\lambda_i(\Lambda_q)} \right) \left( 1 + \frac{2M - \lambda_{min}(\Lambda_q) + \rho}{\lambda_i(\Lambda_q)} \right)} \right) \tag{92}$$

where the final result is due to eq. 59.

$\square$

## B.1 Discussion of the bound

In the following we want to give an insight of how the upper bound in Thm. 4.1 becomes 0 when $N \to \infty$. From the definition of $\sigma_1 = \sigma_2 = \sigma \frac{2C}{N} \to 0$ as $N \to \infty$. Recall

$$\mathbb{E}_{\mathbf{e}} \mathbb{E}_{\mathbf{E}} (D_{kl}[q||p])$$
$$\leq \frac{1}{2} \Bigg[ \underbrace{Tr((-\lambda_{min}(\Lambda_q) + \sqrt{2v(\mathbf{E})\log(d)} + \rho)\Lambda_q^{-1})}_{\text{Term 1}}$$

$$+ \underbrace{\sigma_1^2 \lambda_{max}(\Sigma_q) \sum_{i=1}^{d} \left( \frac{h + H - 1 + \frac{\lambda_{min}(\Lambda_q) + \sqrt{2v(\mathbf{E})\log(d)} - \rho}{\lambda_i(\Lambda_q)}}{hH} \right)}_{\text{Term 2}} \tag{93}$$

$$+ \underbrace{\sum_{i=1}^{d} b_i^2 \left( \frac{h + H - 1 + \frac{\lambda_{min}(\Lambda_q) + \sqrt{2v(\mathbf{E})\log(d)} - \rho}{\lambda_i(\Lambda_q)}}{hH} \right)}_{\text{Term 3}} \underbrace{- \eta_q^\top \Lambda_q^{-1} \eta_q}_{\text{Term 4}}$$

$$+ \underbrace{\mathbf{a}\mathbf{a}^\top \left( -\lambda_{min}(\Lambda_q) + \sqrt{2v(\mathbf{E})\log(d)} + \rho \right)}_{\text{Term 5}} + \underbrace{\sum_{i=1}^{d} \log \left( \frac{h + H - 1 + \frac{\lambda_{min}(\Lambda_q) + \sqrt{2v(\mathbf{E})\log(d)} - \rho}{\lambda_i(\Lambda_q)}}{hH} \right)}_{\text{Term 6}} \Bigg]$$

where $b_i := (\Lambda_q^{\frac{-1}{2}} \eta_q)_i$, $\mathbf{a} := \Lambda_q^{-1} \eta_q$, $h = 1 + \frac{-2M - \lambda_{min}(\Lambda_q) + \rho}{\lambda_i(\Lambda_q)}$ and $H = 1 + \frac{2M - \lambda_{min}(\Lambda_q) + \rho}{\lambda_i(\Lambda_q)}$.

- Term 1:

$$Tr((-\lambda_{\min}(\Lambda_q) + \sqrt{2v(\mathbf{E})\log(d)} + \rho)\Lambda_q^{-1}) \tag{94}$$

$$= Tr((-\lambda_{\min}(\Lambda_q) + \rho)\Lambda_q^{-1}), \text{ due to } \lim_{\sigma_2 \to 0} v(\mathbf{E}) = 0 \tag{95}$$

$$= 0, \text{ because } A = 0 \text{ and } \rho = \lambda_{\min}(\Lambda_q) \tag{96}$$

- Term 2:

$$\sigma_1^2 \lambda_{max}(\Sigma_q) \sum_{i=1}^{d} \left( \frac{h + H - 1 + \frac{\lambda_{min}(\Lambda_q) + \sqrt{2v(\mathbf{E})\log(d)} - \rho}{\lambda_i(\Lambda_q)}}{hH} \right) = 0, \text{ due to } \sigma_1 \to 0 \tag{97}$$

- Term 3:

$$\sum_{i=1}^{d} b_i^2 \left( \frac{h + H - 1 + \frac{\lambda_{min}(\Lambda_q) + \sqrt{2v(\mathbf{E})\log(d)} - \rho}{\lambda_i(\Lambda_q)}}{hH} \right) \tag{98}$$

$$= \sum_{i=1}^{d} b_i^2 \left( \frac{h + H - 1 + \frac{\lambda_{min}(\Lambda_q) - \rho}{\lambda_i(\Lambda_q)}}{hH} \right), \text{ due to } \lim_{\sigma_2 \to 0} v(\mathbf{E}) = 0 \tag{99}$$

$$= \sum_{i=1}^{d} b_i^2 \left( \frac{h + H - 1}{hH} \right), \text{ because } \rho = \lambda_{\min}(\Lambda_q) \tag{100}$$

$$= \eta_q^\top \Lambda_q^{-1} \eta_q, \text{ because } h = H = 1 \text{ and the definition of } b_i \tag{101}$$

$$\tag{102}$$

- Term 5:

$$\mathbf{a}\mathbf{a}^\top \left( -\lambda_{min}(\Lambda_q) + \sqrt{2v(\mathbf{E})\log(d)} + \rho \right) \tag{103}$$

$$= \mathbf{a}\mathbf{a}^\top \left( -\lambda_{min}(\Lambda_q) + \rho \right), \text{ due to } \lim_{\sigma_2 \to 0} v(\mathbf{E}) = 0 \tag{104}$$

$$= 0, \text{ because } \rho = \lambda_{\min}(\Lambda_q) \tag{105}$$

- Term 6:

$$\sum_{i=1}^{d} \log \left( \frac{h + H - 1 + \frac{\lambda_{min}(\Lambda_q) + \sqrt{2v(\mathbf{E})\log(d)} - \rho}{\lambda_i(\Lambda_q)}}{hH} \right) \tag{106}$$

$$= \sum_{i=1}^{d} \log \left( \frac{h + H - 1 + \frac{\lambda_{min}(\Lambda_q) - \rho}{\lambda_i(\Lambda_q)}}{hH} \right), \text{ due to } \lim_{\sigma_2 \to 0} v(\mathbf{E}) = 0 \tag{107}$$

$$= \sum_{i=1}^{d} \log \left( \frac{h + H - 1}{hH} \right), \text{ because } \rho = \lambda_{\min}(\Lambda_q) \tag{108}$$

$$= 0, \text{ because } h = H = 1 \tag{109}$$

## C   Point estimates

We also show the point estimate (through the usual stochastic gradient descent) in Table 5, which helps gauging how well these approximate inference methods are performing in an absolute sense.

Table 5: RMSE on test data (UCI datasets)

| Dataset | Avg. Test RMSE and Std. | |
| | Point Estimate | DP-SEP |
| --- | --- | --- |
| Naval | **0.001 $\pm$ 0.0001** | $0.002 \pm 0.0003$ |
| Kin8nm | $0.091 \pm 0.0015$ | **0.078 $\pm$ 0.0022** |
| Power | $4.182 \pm 0.0402$ | **4.032 $\pm$ 0.1385** |
| Wine | $0.645 \pm 0.0098$ | **0.627 $\pm$ 0.0362** |
| Protein | **4.539 $\pm$ 0.0288** | $4.585 \pm 0.0589$ |
| Year | $8.932 \pm$ NA | **8.862 $\pm$ NA** |

## D  Differentially private variational inference under the neural network model

Similar to the idea of DP-VI, we derive the variational lower bound under the neural network model, and then apply DP-SGD to ensure the approximate posterior to be differentially private.

For simplicity, we treat the noise precision $\gamma$ and the prior precision $\lambda$ as hyperparameters, and we impose priors only on the weights:

$$p(W_0) = \prod_{i=1}^{h_d} \mathcal{N}(\mathbf{w}_{0,i}|0, \lambda^{-1}I), \tag{110}$$

$$p(\mathbf{w}_1) = \mathcal{N}(\mathbf{w}_1|0, \lambda^{-1}I), \tag{111}$$

$$\tag{112}$$

The approximate posterior over the model parameters $\boldsymbol{\theta} = \{W_0, \mathbf{w}_1\}$ is given by

$$q(\boldsymbol{\theta}) = q(W_0)q(\mathbf{w}_1) = \prod_{i=1}^{h_d} \mathcal{N}(\mathbf{w}_{0,i}|\mathbf{m}_i, diag(\mathbf{v}_i))\mathcal{N}(\mathbf{w}_1|\mathbf{m}', diag(\mathbf{v}')) \tag{113}$$

where $\{\mathbf{m}_i, \mathbf{v}_i, m'_k, v'_k\}$ are posterior parameters. Recall from Sec. 6.2, the likelihood of the mini-batch of the data under the neural network model is given by

$$p(\mathcal{D}|\boldsymbol{\theta}) := p(\mathbf{y}|W_0, \mathbf{w}_1, X_s, \gamma) = \prod_{n=1}^{s} \mathcal{N}(y_n|z_{1,n}, \gamma^{-1}) \tag{114}$$

where $\gamma$ is the noise precision and $z_{1,n} = \mathbf{w}_1^\top \sigma(W_0 \mathbf{x}_n)$.

The variational lower bound in this case can be written as

$$\mathbb{E}_{q(\boldsymbol{\theta})} \left[\log p(\mathcal{D}|\boldsymbol{\theta})\right] - D_{KL}[q(\boldsymbol{\theta})||p(\boldsymbol{\theta})]. \tag{115}$$

Given the factorizing Gaussian prior and posterior pairs, the KL divergence is closed form

$$D_{KL}[q(\boldsymbol{\theta})||p(\boldsymbol{\theta})] = D_{KL}[q(W_0)||p(W_0)] + D_{KL}[q(\mathbf{w}_1)||p(\mathbf{w}_1)], \tag{116}$$

$$= \sum_{i=1}^{h_d} \frac{1}{2} \left[ \sum_{j=1}^{d+1} \lambda v_{ij} + \lambda \mathbf{m}_i^\top \mathbf{m}_i - (d+1) - (d+1)\log_e(\lambda) - \sum_{j=1}^{d+1} \log_e (v_{ij}) \right]$$

$$+ \frac{1}{2} \left[ \sum_{i=1}^{h_d} \lambda v'_i + \lambda {\mathbf{m}'_i}^\top \mathbf{m}'_i - h_d - h_d \log_e(\lambda) - \sum_{i=1}^{h_d} \log_e (v'_i) \right]_d \tag{117}$$

We can further expand the first term in eq. 115

$$\mathbb{E}_{q(\boldsymbol{\theta})}\left[\log p(\mathcal{D}|\boldsymbol{\theta})\right]$$

$$= \sum_n \int q(W_0)q(\mathbf{w_1}) \log \mathcal{N}(y_n|\mathbf{w_1}^\top \sigma(W_0\mathbf{x}_n), \gamma^{-1}) \, dW_0 \, d_{\mathbf{w_1}}, \tag{118}$$

$$= \sum_n \int q(W_0) \left[\int \mathcal{N}(\mathbf{w_1}|\mathbf{m}', \mathbf{v}') \left\{-\frac{\gamma}{2}(y_n - \mathbf{w_1}^\top \mathbf{z}_{0,n})^2 - \frac{1}{2}\log|2\pi\gamma^{-1}|\right\}\right] \, dW_0, \tag{119}$$

$$= \sum_n \int q(W_0) \left[-\frac{\gamma}{2}\left(y_n^2 - 2y_n\mathbf{m}'^\top \mathbf{z}_{0,n} + Tr[(\mathbf{z}_{0,n}\mathbf{z}_{0,n}^\top)V' + \mathbf{m}'^\top(\mathbf{z}_{0,n}\mathbf{z}_{0,n}^\top)\mathbf{m}') - \frac{1}{2}\log|2\pi\gamma^{-1}|\right] \, dW_0 \tag{120}$$

where $\mathbf{z}_{0,n} = \sigma(W_0\mathbf{x}_n)$ and $V'$ is a diagonal matrix where diagonal entries are $\mathbf{v}'$. Due to the nonlinearity, the final integral is not analytically tractable. We use the Monte Carlo approximation to the integral by using $L$ samples drawn from $W_0^{(l)} \sim q(W_0)$:

$$\approx \frac{1}{L}\sum_{l=1}^{L}\sum_n \left[-\frac{\gamma}{2}\left(y_n^2 - 2y_n\mathbf{m}'^\top\sigma(W_0^{(l)}\mathbf{x}_n) + Tr[(\sigma(W_0^{(l)}\mathbf{x}_n)\sigma(W_0^{(l)}\mathbf{x}_n)^\top)V'] + \mathbf{m}'^\top(\sigma(W_0^{(l)}\mathbf{x}_n)\sigma(W_0^{(l)}\mathbf{x}_n)^\top)\mathbf{m}')\right]$$
$$- \frac{n}{2}\log|2\pi\gamma^{-1}|. \tag{121}$$

Applying DP-SGD to this objective function requires the sample-wise gradient to be limited by a clipping threshold $C$ and then adding the Gaussian noise with a noise scale tuned by $\sigma C$, where $\sigma$ is the privacy parameter. To compose privacy loss, we use the analytic moments accountant by (Wang et al., 2019).

### D.1 Bayesian neural networks hyper-parameter settings

Here we give a detailed hyper-parameter setting used for computing the VI method and the privatized version under neural networks experiments on the UCI datasets. Table 6 and Table 7 reflects the number of epochs, batch size, learning rate, noise precision ($\gamma$), prior precision ($\lambda$), number of samples used in Monte Carlo approximation, KL-divergence regularizing parameter ($\beta$), clipping threshold ($C$) and the standard deviation for the Gaussian noise ($\sigma$) used for each dataset.

Table 6: VI hyperparameter settings

|         | epochs | batch size | learning rate | $\gamma$ | $\lambda$ | MC samples | $\beta$ |
|---------|--------|------------|---------------|----------|-----------|------------|---------|
| Naval   | 100    | 100        | $1 \cdot 10^{-4}$ | 20   | 1    | 10 | 1     |
| Kin8nm  | 100    | 100        | $1 \cdot 10^{-3}$ | 10   | 100  | 20 | 0.01  |
| Power   | 200    | 100        | $9 \cdot 10^{-4}$ | 18   | 100  | 10 | 0.001 |
| Wine    | 40     | 100        | $1 \cdot 10^{-2}$ | 2    | 200  | 20 | 0.01  |
| Protein | 100    | 200        | $1 \cdot 10^{-5}$ | 50   | 100  | 20 | 2     |
| Year    | 40     | 2000       | $1 \cdot 10^{-5}$ | 1.5  | 1    | 20 | 0.1   |

Table 7: DP-VI hyperparameter settings with a privacy budget $\epsilon \approx 1$ and $\delta = 10^{-5}$

|         | epochs | batch size | learning rate | $\gamma$ | $\lambda$ | MC samples | $\beta$ | $C$ | $\sigma$ |
|---------|--------|------------|---------------|----------|-----------|------------|---------|-----|----------|
| Naval   | 100    | 100        | $1 \cdot 10^{-4}$ | 20   | 1    | 10 | 1     | 5 | 1.3 |
| Kin8nm  | 100    | 100        | $1 \cdot 10^{-3}$ | 10   | 100  | 20 | 0.01  | 1 | 1.7 |
| Power   | 200    | 100        | $9 \cdot 10^{-4}$ | 18   | 100  | 20 | 0.001 | 1 | 2   |
| Wine    | 40     | 100        | $1 \cdot 10^{-2}$ | 2    | 200  | 20 | 0.01  | 5 | 30  |
| Protein | 100    | 200        | $1 \cdot 10^{-5}$ | 50   | 100  | 20 | 2     | 1 | 1.2 |
| Year    | 40     | 2000       | $1 \cdot 10^{-5}$ | 1.5  | 1    | 20 | 0.1   | 1 | 1.1 |

# E   Clipping effect on DP-SEP

We study in this section the effect that the clipping threshold $C$ choice has in DP-SEP posterior estimates for a fixed privacy budget of $\epsilon = 1$ and $\delta = 10^{-5}$. Table  8 and Table  9 shows the averaged results and the standard deviation for the RMSE and test log-likelihood obtained for each dataset on 5 independent runs. As the value of $C$ increases, we are letting the natural parameters to retain more information but at the expense of adding higher variations due to the Gaussian noise addition that also depends on the clipping threshold.

Table 8: RMSE on test data (UCI datasets) from DP-SEP at different $C$

| | Avg. Test RMSE and Std. | | | | | |
|---|---|---|---|---|---|---|
| $C$ | Naval | Kin8nm | Power | Wine | Protein | Year |
| 1 | $0.0025 \pm 0.0007$ | $0.079 \pm 0.0020$ | $3.997 \pm 0.0762$ | $0.600 \pm 0.0470$ | $4.601 \pm 0.0485$ | $9.971 \pm$ NA |
| 2 | $0.0026 \pm 0.0006$ | $0.082 \pm 0.0045$ | $4.007 \pm 0.0707$ | $0.605 \pm 0.0469$ | $4.660 \pm 0.0397$ | $10.013 \pm$ NA |
| 3 | $0.0034 \pm 0.0010$ | $0.083 \pm 0.0076$ | $4.014 \pm 0.0817$ | $0.608 \pm 0.0482$ | $4.692 \pm 0.0517$ | $10.025 \pm$ NA |
| 4 | $0.0034 \pm 0.0007$ | $0.085 \pm 0.0061$ | $4.057 \pm 0.0812$ | $0.612 \pm 0.0468$ | $4.746 \pm 0.0872$ | $10.041 \pm$ NA |
| 5 | $0.0038 \pm 0.0005$ | $0.087 \pm 0.0062$ | $4.081 \pm 0.0886$ | $0.619 \pm 0.0428$ | $4.797 \pm 0.1122$ | $10.064 \pm$ NA |
| 10 | $0.0042 \pm 0.0005$ | $0.090 \pm 0.0078$ | $4.153 \pm 0.0622$ | $0.636 \pm 0.0353$ | $4.946 \pm 0.1312$ | $10.108 \pm$ NA |

Table 9: Test log-likelihood (UCI datasets) from DP-SEP at different $C$

| | Avg. Test log-likelihood and Std. | | | | | |
|---|---|---|---|---|---|---|
| $C$ | Naval | Kin8nm | Power | Wine | Protein | Year |
| 1 | $4.545 \pm 0.2675$ | $1.116 \pm 0.0195$ | $-2.805 \pm 0.0173$ | $-0.903 \pm 0.0651$ | $-2.945 \pm 0.0115$ | $-3.974 \pm$ NA |
| 2 | $4.535 \pm 0.2253$ | $1.108 \pm 0.0167$ | $-2.835 \pm 0.0289$ | $-0.907 \pm 0.0651$ | $-2.968 \pm 0.0160$ | $-4.001 \pm$ NA |
| 3 | $4.443 \pm 0.2373$ | $1.098 \pm 0.0196$ | $-2.845 \pm 0.0372$ | $-0.911 \pm 0.0644$ | $-2.984 \pm 0.0227$ | $-4.015 \pm$ NA |
| 4 | $4.430 \pm 0.2895$ | $1.093 \pm 0.0215$ | $-2.862 \pm 0.0812$ | $-0.915 \pm 0.0621$ | $-3.020 \pm 0.0366$ | $-4.026 \pm$ NA |
| 5 | $4.360 \pm 0.2927$ | $1.091 \pm 0.0237$ | $-2.886 \pm 0.0886$ | $-0.926 \pm 0.0549$ | $-3.064 \pm 0.0698$ | $-4.042 \pm$ NA |
| 10 | $4.242 \pm 0.3607$ | $1.069 \pm 0.0352$ | $-2.915 \pm 0.0622$ | $-0.932 \pm 0.0482$ | $-3.096 \pm 0.0881$ | $-4.099 \pm$ NA |

