# OpenReview forum: "Differentially Private Stochastic Expectation Propagation"
_TMLR — Accepted by TMLR_

### Review · Reviewer_X32H · 2022-07-21

**Summary Of Contributions:**

The proposes a  DP Stochastic Expectation Propagation algorithm for computing a posterior distribution for model parameters in Bayesian learning. There are two existing approaches: DP variational inference (Jälkö et al., 2017) and DP expectation maximization (Park et al., 2017). The experiments indicate that the proposed method often improves upon the baseline methods. A theoretical analysis in terms of KL-distance from the posterior given by the non-DP algorithm is carried out.

**Broader Impact Concerns:**

I have no concerns of ethical implications.

**Requested Changes:**

I think the theory part should be reworked carefully (see the comments above). Even if the resulting bound is pessimistic or not even 'a priori' bound (containing some a posteriori quantities), it should be correct. I cannot see how it would be correct now with that diagonal correction matrix A. Thus I would say that the paper requires a major revision.

**Strengths And Weaknesses:**



Strengths:

- Novel idea: DP Stochastic Expectation Propagation does not seem to be considered before.
- Strong experimental evidence, seems to improve upon the results of DP variational inference (Jälkö et al., 2017) and DP expectation maximization (Park et al., 2017).

Weaknesses:

- Clearly the biggest weakness: I think there are issues with the theory part, that part requires corrections. In more detail below:

I have some difficulties reading the theoretical error bound: what is for example $\sigma_2$ ? If I understand correctly $v(E)$ is of the order $O(d^2)$, and since there is already $d \log d$ in the last term, the error bound is $O(d^2 log d)$ which sounds quite a lot.

I cannot follow the reasoning with the decomposition of Def. 4.2, I believe it is not mathematically correct: I think you cannot necessarily shift the spectrum of a Hermitian matrix by adding a diagonal matrix as you describe. Now clearly $\Lambda_q + E$ is Hermitian. Adding a diagonal A as you describe does not necessarily make the matrix non-negative, I think you should have the correct eigenbasis there as well (corresponding to that of $\Lambda_q + E$).

In any case: even if you would use some correction matrix this way, you should somehow bound it to have 'a priori' error bound. Now the matrix A appears in the error bound, and since it is 'a posteriori' information the error bound is 'a posteriori' bound that you evaluate after running the algorithm. And due to the pessimistic dependence on dimension, I suspect it would quickly become useless as dimension d grows.

Theorem 4.1 is "privacy-accuracy trade-off" bound. But where is the privacy, how is the level of privacy reflected by the bound?

You write: "We look forward to extending our private SEP to more large-scale scenarios such as federated learning settings..." Does large-scale mean high-dimensional as well? I have doubts, whether the approach would work as well in higher dimensions. Isn't this indicated by your KL-divergence error bound as well ( $O(d^2 log d)$ ) ?

As there does not seem to be prior work on DP expectation propagation or stochastic expectation propagation, and since otherwise (other than the theoretical analysis) I have no reason to doubt the correctness of the paper, I think the paper meets most of the criteria for publication. However the theoretical part should be corrected.

---

> ### Author Response · Authors · 2022-08-05
> **RE: Review of Paper165 by Reviewer X32H**
>
> Thank you for your valuable comments on our work. Hopefully, our responses will help resolve your concerns.
>
> Thanks for raising the concern on the difficulties of reading the theoretical error bound. To solve this problem we have added a notation table in the manuscript.
>
> We agree with the reviewer comments on the Theorem 4.1. upper bound. In the new version of the manuscript, we redefined the matrix $A$ as a diagonal matrix with entries equal to - minimum eigenvalue of $\Lambda_q + E$ + \rho, where $\rho$ is a small positive constant to ensure that the resulting $\Lambda_p$ is positive definite. We rederived the bound of the error "a posteriori" for the theorem with the new $A$ matrix. However, the resulting bound is still dimension dependent and of the $O(d^2logd)$ since we use the same upper bound for the eigenvalues of random matrix theory as in the previous version. We wonder if the reviewer has any suggestions on how to get rid of $d$ and if there is a better upper bound we could use for the eigenvalues of Gaussian random matrices.
>
> The level of privacy is reflected in the bound through the noise variance for the first natural parameter, ($\sigma_1$), matrix variance statistic, ($v(E)$), and the diagonal matrix $A$ since those are noise dependent terms. The level of privacy reflected by the bound is clear for $\sigma_1$, but in the case of the noise variance for the second natural parameter $\sigma_2$, it is not so clear as it appears in the bound through $v(E)$ and $A$.
>
> A downside of adding the new scaled identity matrix, $A$, to the noisy posterior second natural parameter is that, after the post-processing step, the variance of the posterior $p$ is likely to increase. This is an unfortunate artifact of using the Gaussian mechanism to privatize the second natural parameter matrix.  We leave the following for future work: incorporating other DP mechanisms (e.g., DP-PCA mechanisms) that preserve the positive definiteness of the covariance/precision matrix, which does not require any post-processing step.

---

> > ### Comment · Reviewer_X32H · 2022-08-24
> > **Thank you for replies**
> >
> > Thank you for addressing my concerns, the paper looks more mature now. I have still one small concern:
> >
> > In Def. 4.2 you have the variables $\rho$ which is a random variable (due to randomness of $E$). As far as I see, you cannot choose a finite $\rho$ for which the given condition would hold with probability 1.
> >
> > Then in Thm. 4.1 you take expectation w.r.t. $E$ but on the right hand side there is still $\rho$ (which depends on $E$).
> >
> > Please try to fix this somehow, by taking e.g. "with high probability" $\rho$.
> >
> > Other than that, I am happy with the changes.

---

> > > ### Author Response · Authors · 2022-08-26
> > > **Regarding $\rho$**
> > >
> > > Thanks for raising the point. We will carefully think about this, and update our theorem accordingly. All the authors are on summer vacation at the moment, so there will be some delay in our update, possibly to be done in early/mid September. Thanks for your understanding and patience.

---

> > > ### Author Response · Authors · 2022-09-16
> > > **RE: Regarding $\rho$**
> > >
> > > Due to the way the matrix $A$ is constructed in Def 4.2, we have that the minimum eigenvalue of $\Lambda_p$ is equal to $\rho$, independent of what value $E$ takes. In other words, $\lambda_{\min}(\Lambda_q + E + A) = \lambda_{\min}(\Lambda_q + E + (-\lambda_{\min} (\Lambda_q + E) + \rho)I) =  \lambda_{\min}(\Lambda_q + E) - \lambda_{\min}(\Lambda_q + E ) + \rho = \rho$. This indicates that $\rho$ does not depend on the random variable $E$ and hence, it is not necessary to take expectation over $E$ for $\rho$.
> > >
> > > In contrast, one can construct $A$ by $A = \rho I$, where $\rho>0$, and $\Lambda_p = \Lambda_q +E + \rho I$. To ensure $\Lambda_p$ to be positive definite, we need to set $\rho > \lambda_{\min} (\Lambda_q + E)$. Hence, in this construction, $\rho$ depends on the distribution of $E$. And one can derive a Theorem like ours, but that Theorem will hold with this probability, $P_{E}( \lambda_{\min} (\Lambda_q + E) > - \rho) $.

---

### Review · Reviewer_SHsv · 2022-07-24

**Summary Of Contributions:**

In general, this paper develops the private version of the stochastic expectation of propagation. For the case of EP, the sensitivity analysis is difficult whereas, for SEP, it is straightforward. This paper gives an algorithm as well as an analysis for its implementation. Most importantly, the theoretical analysis on the utility is given.

**Requested Changes:**

I have some questions which are related to the DP-SEP algorithm, 1) in line 7 in alg. 3, why not just add the noise to the clipped result ( it is still DP), will this be any different on the performance? 2) why do you do the clipping in line 12 in alg. 3, if only for ensuring DP, I think it is not necessary.

Maybe due to the nature of SEP, having to pick one data point at one time to do updates, making the algorithm not scalable to large datasets, in contrast, DP-SGD samples a batch of data to do learning, I am wondering if there is a way to circumvent such a challenge


**Strengths And Weaknesses:**

Strength: non-trivial utility result and relative complete comparison is made

Weakness: adopting DP to SEP is trivial and the analysis is straightforward. from the experimental result, the clipping threshold C seems to have a big impact on the performance if not chosen appropriately, some discussion which is missing on choosing C seems necessary.

---

> ### Author Response · Authors · 2022-08-05
> **RE: Review of Paper165 by Reviewer SHsv**
>
> Title
> RE: Review of Paper165 by Reviewer SHsv
>
> Comment
> Thank you for your valuable comments on our work. Hopefully, our responses will help resolve your concerns.
>
> Answer 1: If one considers to add noise to the clipped $f_n (\theta)$, when calculating the sensitivity analysis of the clipped $f_n (\theta)$, we end up with $\Delta_{\theta_{f_n}} \leq 2C$. In this way, the $\gamma/N$ factor in our proposed algorithm is lost, which implies adding a higher amount of noise to the data dependent term.
>
> Answer 2: The main reason for clipping in line 12 in alg. 3 is to ensure that the sensitivity analysis between iterations holds since the approximated factor $f(\theta)$ update is affected by $f_n (\theta)$.
>
> Answer 3: Both EP and SEP are parallelizable algorithms. In the parallelized setting, if a mini-batch of M samples is considered, step 3 and 4 in SEP algorithm are performed in parallel for each datapoint in the mini-batch. The approximating factor then, is updated via $q_{new}(\theta) = q_{-m} (\theta) f(\theta)^{1-M/N} \prod_{m=1}^{M} f_m(\theta)$. In the DP setting, the sensitivity analysis can be carried out by considering without loss of generality that the $M$ datapoint is the one different from both neighboring datasets and we will end up with the same upper bound for this case.

---

> > ### Comment · Reviewer_SHsv · 2022-08-09
> > **Still have some questions for answer 1 and 2**
> >
> > For 1, I think the author's response still does not answer my original question. Looking at question 5, if the noise is added to the clipped term right after line 7 in Algorithm 3, the factor $\gamma/N$ still gets reserved as the noise will get multiplied by $\gamma/N$ just as the clipped term itself. It's not that the factor will be lost if noise is added to the clipped term right after line 7 in Algorithm 3.
> >
> > For 2, the sensitivity analysis (Proof of Proposition 1) all depends on the clipping operation in line 7 of Algorithm 3, how does the operation in line 12 affect the sensitivity analysis?

---

> > > ### Author Response · Authors · 2022-08-09
> > > **Our answer to Q1**
> > >
> > > This is a very good point. Sorry for our earlier answer not explaining this point well.
> > >
> > > You can certainly add noise to the line 7 in Alg 3, and if that factor itself is DP, \theta_new is DP, as f(theta) is from the previous step and also DP (and  $\\theta_0$ is data-independent anyway). And you're also right about $\\gamma/N$ remaining the same if we do that, due to the scaling happening in eq.(5).
> > >
> > > However, there is a subtle difference between these two cases: (1) adding noise to the line 7; versus (2) adding noise to eq.(5). If we add noise to the line 7, only $\\theta_{f_n}$ is affected by the noise, while the terms, $\\theta_0$ and $\\theta_f$, are unaffected. On the other hand, if we add noise to eq.(5), this noise is spread across all three terms: $\\theta_{f_n}$, $\\theta_0$, $\\theta_f$. Because the noise level is the same in both cases, we expect the accuracy of the DP-SEP posterior would be similar. However, this might cause different behaviours in terms of stability of DP-SEP training. We will empirically investigate this in the future.

---

> > > ### Author Response · Authors · 2022-08-09
> > > **Our answer to Q2**
> > >
> > > We acknowledge that our earlier answer was not explaining this point well.
> > >
> > > Yes, if you look at the proof of Proposition 1, indeed the sensitivity is independent of the norm $\\theta_f$, as long as the two parties having neighbouring datasets start with the same $\\theta_f$ and $\\theta_0$.
> > >
> > > The necessity of normalization of $\\theta_f$ is subtle but important. In eq.(5), by removing $1/N$ factor and setting the damping factor $\\gamma=1$ for simplicity, we arrive at $\\theta_{new} \\leftarrow \\theta_{f_n} + (N-1) \\theta_{f} + \\theta_{0}$.  By normalizing $\\theta_{f}$ and $\\theta_{0}$ by $C$, we ensure that all three factors $\\theta_{f_n}$, $\\theta_{f}$, and $\\theta_{0}$ have the same strength in terms of their norms. This allows us to be consistent with the core reasoning in the construction of the SEP algorithm, i.e., a global factor $\\theta_{f}$ representing an average contribution of the likelihood term to the posterior. In other words, by normalizing $\\theta_{f}$ by C, we ensure that $\\theta_{f_n}$'s contribution to posterior and that of $\\theta_{f}$ match in terms of their norms, as both of them model a single data point's contribution to the posterior distribution. Let us know if this explanation makes sense to the reviewer. If so, we will add this point to the manuscript.

---

> ### Author Response · Authors · 2022-08-09
> **not scalable to large datasets**
>
> There is a parallel version of SEP algorithm to tackle large datasets. See section 4.1 in the [SEP paper](https://arxiv.org/pdf/1506.04132.pdf) for details. One can also use more than one global approximating factor, as in the distributed SEP algorithm (Section 4.3).  We did not incorporate these versions of SEP in our algorithm, although such an extension can be done in future work.

---

### Review · Reviewer_m1oy · 2022-07-25

**Summary Of Contributions:**

This paper studies private approximate Bayesian inference and proposes a DP-SEP algorithm based on SEP. The privatization method is similar to DPSGD (with some differences). From the theoretical perspective, this paper provides sensitivity analysis, differential privacy guarantee, and accuracy guarantee. The paper also provides experimental studies on two tasks: mixture of Gaussians for clustering and Bayesian neural network.

**Broader Impact Concerns:**

No.

**Requested Changes:**

1. Include the explicit expression of the relation between \epsilon, \delta, and \sigma in proposition 2.
2. Discussion Theorem in more detail (like what Theorem 4.1 implies what/where each term is from/from Theorem 4.1, what can we say about the overall accuracy).
3. Address my concerns about the experiments.
4. To make this paper stronger, it might be interesting to empirically study the choice of clipping norm and clipping as a form of regularization.

**Strengths And Weaknesses:**

Strengths: SEP provides better uncertainty estimates than variational inference. Having a private version of SEP is useful for private Bayesian inference. The theoretical results are sound to me. The experimental results provide an extensive comparison with the non-private baseline and other private algorithms.

Weaknesses: I think the novelty of this paper is not significant enough. Although the paper lists a few differences compared to DPSGD. I think the only difference is to not use the batch (which is not from a privacy perspective.) And clipping for statistics that have unbounded domain is widely used in many privacy + statistics papers. The privacy guarantee follows immediately from previous results. The paper even doesn't provide the relation between \epsilon, \delta, and \sigma. Even if they use the auto-dp package, I think proposition 2 should include the explicit expression or at least the order of this relation. As for the accuracy guarantee, Theorem 4.1 is only for per round, and it doesn't provide any formal guarantee for the overall accuracy. Moreover, Theorem 4.1 is not that informative, and more explanation should be discussed. For the experiments, I also have two concerns. 1. In Table 1, it seems that DP-SEP is much worse than DP-EM under a reasonable \epsilon. 2. The paper claims that ``as we increase the privacy loss, the performance of DP-SEP gets closer to the non-private ones and the ground truth.'' on page 10. However, from Figure 1, it seems that the performance of \epsilon=50 is almost the same as \epsilon=5.

---

> ### Author Response · Authors · 2022-08-05
> **RE: Review of Paper165 by Reviewer m1oy**
>
> Thank you for your valuable comments on our work.
>
> Answer 1: Thank you for raising this point. We have added the explicit expression of the relation between $\epsilon$, $\delta$ and $\sigma$ for the Gaussian mechanism in the Differential Privacy section in the new version of the manuscript.
>
> Answer 2: Thanks for the comments and concerns about Theorem 4.1.  As we stated in the manuscript the Theorem is only per round and due to the nested structure of the algorithm it is difficult to derive the result after multiple iterations, thus, talk about overall accuracy. A new version of the Theorem 4.1 and it's proof can be found in the updated version of the manuscript.
>
> Answer 3: 1. Under stronger privacy contraints ($\epsilon=1$), DP-SEP is not capable to learn the mean parameters for the different clusters, and hence, this is reflected in the F-norm on the mean. On the contrary, DP-EM approximation for the mean parameters is  better than DP-SEP but at the cost of a poor approximation for the covariance parameters.
>
> 2. In Figure 1, for $\epsilon=5$ almost all datapoints corresponding to the green cluster are misclassified to the yellow one. Increasing the value of epsilon from 5 to 50 helps to fix the misclassification on those datapoints. Aditionally, results in Table 1 show that the accuracy of the posterior for $\epsilon=50$ is close to clipped SEP with $C=1$ while for $\epsilon=5$ this is not the case.
>
> Answer 4: We agree with the reviewer suggestion about the interest in empirically study the choice of $C$. In Table 8 and Table 9, in Section E, we show the clipping norm effect on PBP experiment for different $C$ values.

---

> > ### Comment · Reviewer_m1oy · 2022-09-09
> > **Response**
> >
> > Thank you for answering my questions. I don't think the authors fully answered my questions.
> >
> > 1. I was asking the overall \eps \delta expression after you do privacy accountant, not just for the Gaussian mechanism.
> > 2. The new version of Theorem 4.1 looks better, but it's still hard to parse it. In particular, it's not straightforward to see the gap gets closed when \sigma_1 and \sigma_2 go to 0. Section B.1 does not provide enough details.
> >
> > Minor comments:
> >
> > 1. There are a few missing punctuation marks before and in Theorem 4.1.
> > 2. Section B.1 ``in the no noise scenario we can A → 0.''

---

> > > ### Author Response · Authors · 2022-09-16
> > > **Re: Include the explicit expression of the relation between \epsilon, \delta, and \sigma in proposition 2.**
> > >
> > > Unfortunately, there is no closed form expression for what the reviewer requests. But we can provide an overview on how this can be computed. The composition we used is the composition of subsampled R\'enyi differential privacy (RDP), where the RDP upper bound of a randomized mechanism on a subsampled data is determined by Theorem 9 in [Wang et al 2019](https://arxiv.org/pdf/1808.00087.pdf). Then, this expression is multiplied by how many times we repeat the SEP steps. And the overall RDP upper bound is converted to DP using Lemma 6 in [Wang et al 2019](https://arxiv.org/pdf/1808.00087.pdf). We used the [autodp](https://github.com/yuxiangw/autodp) package (which is based on Wang et al 2019) to compute the corresponding $\sigma$ given $\epsilon$ and $\delta$.

---

> > > ### Author Response · Authors · 2022-09-16
> > > **Re: not straightforward to see the gap gets closed when \sigma_1 and \sigma_2 go to 0. Section B.1 does not provide enough details.**
> > >
> > > We updated our manuscript to incorporate this point. Please take a look at the new Section B.1 of the uploaded manuscript.

---

### Decision · Action_Editors · 2022-10-09

**Recommendation:** Accept as is

**Comment:**

The paper improved substantially during the reviewing process and all reviewers are happy with the updated version of the paper. The main remaining criticism is that there isn't an end-to-end utility analysis. I decided that since most Bayesian approaches, even without privacy guarantees, do not have a theoretical performance guarantee, it is acceptable that the current paper evaluates their methods empirically. All reviewers agree that the empirical results are of high quality. Moreover, the new approach for private Bayesian learning (returning an approximate posterior distribution) gives good utility at reasonable DP parameters.

That said, if the authors could still somehow analyze the approximation of the private posterior distribution vs its non-private counterpart (notice that this does not require the non-private version of the algorithm to have theoretical guarantees), it would increase the technical contribution of the work. This could be left as a future work if the authors find it on-trivial to state and prove a utility guarantee of this kind.

The authors should incorporate the reviewer's suggestions and the paper is ready to publish (without having another round of reviews).

Congratulations to the authors!

**Audience:**

The targeted audience of the paper is researchers and practitioners at the intersection of differential privacy and machine learning.

**Claims And Evidence:**

The paper claims that the proposed algorithm satisfies differential privacy. It is accurate, convincing, and standard in the context of literature.  The paper also reports strong empirical results which demonstrate the merits of the method which improves over existing differentially private Bayesian learning.